

# A prognostic pollen emissions model for climate models (PECM1.0)

Matthew C. Wozniak[1], Allison Steiner[1]

[1]Climate and Space Sciences and Engineering, University of Michigan, Ann Arbor, MI 48109, USA

*Correspondence to*: Matthew C. Wozniak (mcwoz@umich.edu)

**Abstract.** We develop a prognostic model of Pollen Emissions for Climate Models (PECM) for use within regional and global climate models to simulate pollen counts over the seasonal cycle based on geography, vegetation type and meteorological parameters. Using modern surface pollen count data, empirical relationships between prior-year annual average temperature and pollen season start dates and end dates are developed for deciduous broadleaf trees (*Acer, Alnus, Betula, Fraxinus, Morus, Platanus, Populus, Quercus, Ulmus*), evergreen needleleaf trees (Cupressaceae, Pinaceae), grasses (Poaceae; $C_3$, $C_4$), and ragweed (*Ambrosia*). This regression model explains as much as 57% of the variance in pollen phenological dates, and it is used to create a "climate-flexible" phenology that can be used to study the response of wind-driven pollen emissions to climate change. The emissions model is evaluated in a regional climate model (RegCM4) over the continental United States by prescribing an emission potential from PECM and transporting pollen as aerosol tracers. We evaluate two different pollen emissions scenarios in the model: (1) using a taxa-specific land cover database, phenology and emission potential, and (2) a PFT-based land cover, phenology and emission potential. The resulting surface concentrations for both simulations are evaluated against observed surface pollen counts in five climatic subregions. Given prescribed pollen emissions, the RegCM4 simulates observed concentrations within an order of magnitude, although the performance of the simulations in any subregion is strongly related to the land cover representation and the number of observation sites used to create the empirical phenological relationship. The taxa-based model provides a better representation of the phenology of tree-based pollen counts than the PFT-based model, however we note that the PFT-based version provides a useful and "climate-flexible" emissions model for the general representation of the pollen phenology over the United States.



## 1 Introduction

Pollen grains are released from plants to transmit the male genetic material for reproduction. When lofted into the
atmosphere, they represent a natural source of coarse atmospheric aerosols, ranging from 10 to 70 µm in diameter.
In the mid-latitudes, much of the vegetation relies dominantly on anemophilous, or wind-driven, pollination [*Lewis*
*et al.*, 1983], representing a closely coupled relationship of pollen emissions to weather and climate. Anemophilous
pollinators include woody plants such as trees and shrubs, as well as other non-woody vascular plants such as
grasses and herbs. Pollen emissions are directly affected by meteorological (e.g., temperature, wind, relative
humidity) and climatological (e.g., temperature, soil moisture) factors [*Weber*, 2003]. Aerobiology studies indicate
that after release, pollen can be transported on the order of ten to a thousand kilometers [*Schueler and Schlünzen*,
2006; *Sofiev et al.*, 2006; *Kuparinen et al.*, 2007] but there are still large uncertainties regarding emissions and
transport of pollen.
Prognostic pollen emissions are useful for the scientific community and public, specifically for forecasting
allergenic conditions or predicting the flow of genetic material. To date, most pollen emissions models focus on
relatively short, seasonal time scales and smaller locales for a limited selection of taxa [*Sofiev et al.*, 2013; *Zhang et*
*al.*, 2014; *Liu et al.*, 2016]. Climatic changes in large-scale pollen distributions are mostly absent from scientific
literature, though multiple studies on phenological changes in the pollen season have been published [*Yue et al.*,
2015; *Zhang et al.*, 2015]. In contrast to most meteorological pollen models, climate models require long-term (e.g.,
decadal to century scale) emissions at a range of resolutions covering regions like continents up to the global scale.
This distinction in both time and space requires a flexible model that can account for emissions without taxon-
specific emission data (i.e. differentiation between genera or species) and can be used within aggregated vegetation
descriptions, such as plant functional types. Given recent interest in biological particles and their role in climate
[*Despres et al.*, 2012; *Myriokefalitakis et al.*, 2017], an emissions model that captures longer temporal scales and
broader spatial scales is key to developing global inventories and understanding pollen's role in the climate system.
Here we develop a model for use in the climate modeling community that can be used to simulate pollen emissions
on the decadal or centurial time scale.
Existing pollen forecasting models are often classified as either process-based phenological models or observation-
based models [*Scheifinger et al.*, 2013]. Process-based phenological models employ a parameterization of plant
physiology and climatic conditions (e.g., relating the timing of flowering to a chilling period, photoperiod, or water
availability). Pollen season phenology in an anemophilous species is inherently connected to its environment via
relationships in the growing season dynamics (e.g. bud burst and temperature, [*Fu et al.*, 2012]), and many models
apply the same techniques to flowering as for bud burst [*Chuine et al.*, 1999]. This approach to phenology could be
suited to climate models, given its flexibility for adaptive phenological events and regional-scale studies. Typically,
these types of phenological models are taxa specific as well as regionally dependent, e.g., *Betula* in Europe or
ragweed in California [*Siljamo et al.*, 2013; *Sofiev et al.*, 2013; *Zhang et al.*, 2014]. These models are usually
calibrated to local data only even though distinct geographic differences exist for pollen phenology. Thus, such
models may not perform equally well in other locations. Though process-based models draw a connection between
an atmospheric state variable, i.e. temperature, and pollen emissions, at least three parameters are required for



optimization and they are susceptible to overfitting [*Linkosalo et al.*, 2008]. While some process-based models may
be scaled up to larger regions while maintaining appreciable accuracy [*García-Mozo et al.*, 2009],  such models are
generally not practical for implementation in larger-scale climate modeling with regional climate models (RCMs)
and global climate models (GCMs) because sufficient land cover data is not available at the appropriate taxonomic
level.
In contrast to process-based models, observation-based methods determine the phenology of vegetation with
statistical-empirical approaches (e.g., relating the start of the pollen season with mean temperatures preceding the
pollen event) and often rely on regression models or time series modeling [*Scheifinger et al.*, 2013]. Time series
modeling utilizes observations to define the deterministic and stochastic variability of pollen count observations and
is frequently used in aerobiological studies [*Moseholm et al.*, 1987; *Box et al.*, 1994]. Regression models, either
using a single or multiple explanatory variable(s), exploit past relationships to define the magnitude of emissions as
well as timing variables such as the start date and duration of the pollen season [*Emberlin et al.*, 1999; *Smith and*
*Emberlin*, 2005; *Galán et al.*, 2008]. Using local pollen count data, *Zhang et al.* [2014] completed a regional
phenological analysis using multiple linear regressions for pollen in Southern California for six taxa. *Olsson and*
*Jönsson* [2014] show that empirical models based solely on spring temperature perform just as well as process-based
models using the temperature forcing concept, and better than those including a chilling or dormancy-breaking
requirement.
Observation-based methods assume stationarity, or the likelihood that the statistics of pollen counts or climate
variables are not changing over time.  For these models to apply outside of calibration period, they require that the
driving pattern or relationship is maintained in the future (or past).  For example, as the Earth's climate changes,
these models do not represent the complex connections between pollen emissions and a warming world aside from
the relationships determined empirically. However, these models provide clear and often simple formulations that
have predictable behaviors and forgo the nuance of fitting ambiguous and uncertain parameters. We therefore
choose to employ elements of the observational methods for this pollen emissions model formulation, as described
in Section 4.
In addition to understanding the release of pollen grains, a second consideration is the large-scale transport of pollen.
Once emitted to the atmosphere, pollen is mixed within the atmospheric boundary layer by turbulence, and
depending on large-scale conditions, can be transported far from the emission source.  Prior studies have used both
Lagrangian [*Hidalgo et al.*, 2002; *Hunt et al.*, 2002] and Eulerian techniques to simulate the transport of pollen, with
the former typically used for studies of crop germination and the latter primarily for allergen forecasting. For
example, *Helbig et al.* [2004] used the meteorological model KAMM (Karlsruher Meteorologisches Modell) with
the DRAIS (Dreidimensionales Ausbreitungs- und Immissions-Simulationsmodell) turbulence component to
simulate daily pollen counts for region over Europe. *Schueler and Schlünzen* [2006] use a mesoscale atmospheric
model (METRAS) to quantify the release, transport and deposition of oak pollen for a two-day period over Europe.
*Sofiev et al.* [2013] includes the long-range transport of birch pollen over Western Europe by developing a birch
pollen map and a flowering model to trigger release in the Finnish emergency modeling system (SILAM). *Efstathiou*
*et al.* [2011] developed a pollen emissions model for use within the regional air quality model (the Community



Multi-scale Air Quality model (CMAQ)), and tested their model with birch and ragweed taxa. *Zhang et al.* [2014]
implements a similar pollen emissions scheme with a regional numerical weather prediction model (the Weather
Research and Forecasting (WRF) modeling system). *Zink et al.* [2013] developed a generic pollen modeling
parameterization for use with a numerical weather prediction model (COSMO-ART) that is flexible to include
differing pollen taxa. Collectively, these relatively new developments suggest a growing interest in the prognostic
estimation of pollen on the short-term for seasonal allergen forecasting on the weather (e.g., one to two weeks) time
scale.
In this manuscript, we build on these coupled emissions-transport models and develop a comprehensive emissions
model (Pollen Emissions for Climate Models; PECM) for use at climate model time scales that covers the majority
of pollen sources in sub-tropical to temperate climes, including woody plants, grasses and ragweed. First, we
summarize the spatial distribution and seasonality of pollen counts for various taxa in the United States based on
current observations (Section 2). Then we develop new pollen emissions parameterization for climate studies
(Section 4), transport these emissions over the continental United States (CONUS) using the Regional Climate
Model version 4 (RegCM4) [*Giorgi et al.*, 2012], and evaluate the results using eight years of observed pollen count
data (Section 5). We implement two different land cover classification schemes to illustrate the uncertainties
associated with vegetation representation for trees including: (1) detailed family- or genus- level tree distributions
over CONUS, and (2) the use of plant functional type (PFT) level distributions, which groups vegetation types by
physiological characteristics (Section 3). As the latter provides a greater opportunity for expansion into regional and
global scale climate models over multiple domains, we discuss the effects that the PFT-based categorization has on
the total estimated source strength of pollen. Finally, the limitations of this emissions framework and suggestions for
future developments are included (Section 6).

## 2 Observed pollen Phenology

### 2.1 Data description

The National Allergy Bureau (NAB) of the American Academy of Allergy, Asthma and Immunology (AAAAI)
conducts daily pollen counts at 96 sites in cities across the United States (US), its territories and several locations in
southern Canada (Table S1). All NAB sites implement a volumetric air sampler and certified pollen count experts to
conduct daily pollen counts (grains m$^{-3}$) for up to 42 plant taxa at either the family level (e.g., Cupressaceae,
Poaceae), genus level (e.g., *Acer*, *Quercus*), or for four generic categories termed "Other Grass Pollen," "Other Tree
Pollen," "Other Weed Pollen" or "Unidentified." We use NAB pollen count data ranging from 2003-2010 at all
stations in the continental United States (Figure 1; Table 1) for selected taxa to develop and evaluate PECM, and to
determine the phenology of wind-driven pollen. Individual station locations and descriptions are included in Table
S1.
We evaluate the observed pollen counts to determine the vegetation types that emit the largest magnitude of pollen
over the continental United States. Since many of the taxa reported at the 96 NAB sites frequently have very low



pollen counts (e.g., less than 10 grains m$^{-3}$), a threshold for the grain count is set to select the taxa with the highest
pollen counts. We calculate the average of the annual maximum pollen count across all years (2003-2010), $P_{avgmax}$,
at each site for each counted taxon. We then select taxa to include in PECM using two criteria: (1) the maximum of
$P_{avgmax}$ among all stations exceeds 100 grains m$^{-3}$, and (2) the average $P_{avgmax}$ among all stations exceeds 70 grains m$^{-}$
$^{3}$ (Table 2). Using these two criteria, 13 taxa are selected for inclusion in the model, including *Acer, Alnus,*
*Ambrosia, Betula,* Cupresseceae, *Fraxinus,* Poaceae*, Morus,* Pinaceae, *Platanus, Populus, Quercus* and *Ulnus.*
These thirteen taxa account for about 77% of the total pollen counted across the United States during 2003-2010.
The 13 dominant pollen types are grouped into four main categories by plant functional type: deciduous broadleaf
forest (DBF), evergreen needle-leaf forest (ENF), grasses (GRA) and ragweed (RAG). Plant functional type is a
land cover classification commonly used in the land surface component of climate models, and this categorization
will allow flexibility to apply the emissions model to other climate models. The DBF category includes 9 genus-
level taxa (*Acer, Alnus, Betula, Fraxinus, Morus, Platanus, Populus, Quercus,* and *Ulmus*) and the ENF category
includes two family-level taxa (Cupressaceae and Pinaceae). The grass PFT utilizes pollen count data from the
Poaceae family, although we note that the grass PFT classification may include herbs and other non-woody species
that may emit pollen as well. *Ambrosia* (ragweed) is segregated as its own category (RAG), due to its high pollen
counts in the early autumn and unique land cover features. Daily pollen counts were summed for each PFT prior to
calculating a climatological average.

**2.2 Observed seasonality of pollen emissions**
Pollen counts are analyzed over five subregions based on their climatic differences (Table S1; Figure 1) to identify
emissions patterns over the continental United States. These five subregions include the Northeast (temperate; north
of 38°N and east of 100°W; 34 stations), the Southeast (temperate, subtropical; south of 38°N and east of 100°W; 29
stations), Mountain (varied climate; 100 and 116°W; 9 stations), California (Mediterranean, varied climate; west of
116°W and south of 40°N; 13 stations) and the Pacific Northwest (temperate rainforest; west of 116°W and north of
40°N; 4 stations). Figure 2 shows the observed climatological PFT daily pollen counts averaged over all stations
within the defined subregions.
For deciduous broadleaf forest (DBF) taxa, the Southeast has the highest climatological pollen maximum reaching
up to about 700-1200 grains m$^{-3}$ around day 100. In the Northeast, DBF is the dominant PFT, reaching up to an
average of 400 grains m$^{-3}$ and peaking slightly later (around day 120) than the Southeast. California sites show a
climatological maximum around 150 grains m$^{-3}$ occurring slightly earlier around day 80. A sharp maximum of 775
grains m$^{-3}$ appears in the Mountain subregion at about day 80, with a secondary emission reaching around 150 grains
m$^{-3}$ on day 125. In the Northwest, DBF pollen has the earliest maximum (day 70) at about the same magnitude as
California (~200 grains m$^{-3}$). In some locations, there is a secondary DBF peak in the late summer and early fall due
to the late flowering of *Ulmus crassifolia* and *Ulmus parvifolia*, located predominantly in the Southeast and
California [*Lewis et al.*, 1983]. In the Southeast this occurs between days 225 and 300, while in California this
occurs twice around day 245 and day 265.



The two ENF families exhibit pollen release at two distinct but overlapping times, with Cupressaceae peaking before
Pinaceae. Cupressaceae in the Southeast emits pollen earlier than in other subregions, with a maxima at just over
400 grains m$^{-3}$ around day 10 and counts above 200 grains m$^{-3}$ in December of the prior year. Cupressaceae
dominates the total emissions for the Southeast, with a smaller maximum from Pinaceae of about 180 grains m$^{-3}$
near day 110. In the Northeast, the bimodality of ENF is evident with the Cupressacaeae family reaching a
maximum of 100 grains m$^{-3}$ near day 85 with a secondary Pinaceae maximum approximately 65 days later at about
half the magnitude (~50 grains m$^{-3}$). In the Mountain and Pacific Northwest subregions, the maximum occurs around
day 50-80 and can reach up to 350 grains m$^{-3}$ in the Mountain subregion, but in both subregions is generally much
lower than the eastern United States (approximately 50 grains m$^{-3}$). In the California subregion, ENF emissions are
comparatively low (< 50 grains m$^{-3}$) which is likely due to the bias in sampling locations.
The grasses (Poaceae) have comparatively low climatological pollen counts (<25 grains m$^{-3}$) throughout the season
in all subregions except the Northwest, where the maximum reaches 75 grains m$^{-3}$. However, the average maximum
Poaceae pollen count at individual stations is close to 100 grains m$^{-3}$, with the individual annual maxima reaching
several hundreds of pollen grains m$^{-3}$. Observations by *Craine et al.* [2011] of Poaceae in a prairie have indicated
that $C_3$ and $C_4$ grass flowering occurs at distinctly different times, with $C_3$ in the late spring and $C_4$ in mid- to late
summer. In the AAAAI data, there are two distinct maxima in the Northeast Poaceae count, and we attribute the first
seasonal maximum to $C_3$ grasses (peak at day 155) and the second grass maximum to $C_4$ grasses (peak at day 250).
Although the $C_3$-$C_4$ separation cannot be confirmed in the pollen count data itself, this distinction is included in the
model as discussed in Sections 3.1 and 4.2 below. In the Southeast, this separation of the Poaeceae pollen counts is
less apparent because both of the emission maxima are broader and intersect one another. In the Southeast, the first
observed pollen maximum (assessed as $C_3$ grass pollen) peaks earlier around day 140, while the second maximum
(assessed as $C_4$ grasses) have a similar, yet smaller value around day 250. In the Mountain subregion, the first grass
maximum occurs later in the year (day 175) and the second grass maximum occurs around day 250 in the late
summer. Pollen counts in California are only substantial during the earlier flowering time (C3 grasses) and have a
similar duration to the Northeast, peaking at around day 135. For the Pacific Northwest, there is one strong early
peak of grass pollen in the middle of the summer (day 170) and a secondary maximum is negligible, although counts
below 10 grains m$^{-3}$ register around days 250-270.
Ragweed (*Ambrosia*) pollen is segregated from other grasses and herbs because of the strong allergic response in
humans to this specific species and the unique timing of emissions. Because it is a short-day plant (i.e. its
phenology driven by a shortening photoperiod and cold temperatures [*Deen et al.*, 1998]), ragweed pollen seasons
are generally constrained to the late summer with the exception of the Mountain region where some counts occur in
the spring. Emissions in the Northeast reach a maximum around day 240 at 60 grains m$^{-3}$ while they occur slightly
later in the Southeast, peaking around day 270 with twice the magnitude (120 grains m$^{-3}$). Ragweed pollen in the
Mountain subregion with an expected peak at around day 245, but also an earlier peak at around day 130 with no
confirmed cause. *Ambrosia* is not detected in the climatological station averages for California and the Pacific
Northwest, although some individual sites in these regions record relatively low counts on the order of 10 grains m$^{-3}$.



**3 Model input data**
**3.1 Land cover data**
With a goal of developing regional to global pollen emissions, one of the greatest limitations is the description of
vegetation at the appropriate taxonomic level and spatial resolution. While land cover databases specific to species
level are available for some regions, they are not available globally. Alternatively, vegetation land cover in regional
to global models can be represented by classifications based on biophysical characteristics. For climate models, a
common approach to represent land cover is with plant functional types (PFT), and global PFT data is readily
available and used by many regional and global climate models to describe a variety of terrestrial emissions
[*Guenther et al.*, 2006] and biophysical processes in land-atmosphere exchange models. The creation of a pollen
emissions model with PFT categorization would be of use at a broad range of spatial scales and domains while
integrating more readily with climate models. In the pollen emissions model development and evaluation (Sections
4 and 5), we compare two different vegetation descriptions of broadleaf deciduous and evergreen needleleaf trees
including (1) family- or genus-specific land cover and (2) land cover categorized by PFT.
The Biogenic Emissions Landuse Database (BELD) provides vegetation species distributions over the continental
United States at 1 km resolution based on land surveys [*Kinnee et al.*, 1997]. The BELD database includes 230
different tree, shrub and crop taxa across the United States as a fraction of the grid cell area at either the genus or
species level. For family and genus level pollen emissions, the BELD land cover fraction for the 11 dominant
pollen-emitting tree taxa identified in Section 2.1 is utilized (Table 1; Figure 3). For species level land cover data,
land cover fraction is calculated as the aggregate of all species within a family or genus.
For the PFT land cover, we use the Community Land Model 4 (CLM4) [*Oleson et al.*, 2010] surface dataset that
employs a 0.05º resolution satellite-derived land cover fraction from the International Geosphere Biosphere
Programme (IGBP) classification [*Lawrence and Chase*, 2007]. We sum all three biome PFT categories (temperate,
tropical and boreal) for deciduous broadleaf forests (DBF) and two biome PFT categories (boreal and temperate) for
evergreen needleaf forests (ENF) to produce the model PFT land cover.
Figures 4a-d compare the BELD land cover (summed by PFT) and CLM4 land cover for the two tree PFTs. An
important distinction is that CLM4 land cover extends beyond U.S. borders because it is derived from a global
dataset, whereas BELD is constrained to the continental United States. BELD and IGBP land cover show general
agreement on the regional distribution of both tree PFTs. DBF is predominantly in the eastern portion of the United
States with a gap in the Midwestern corn belt. ENF is present in the Southeast, the Northeast along the U.S.-
Canadian border, along the Cascade and Coastal mountain ranges and throughout the northern Rockies. A notable
difference is the IGBP representation of ENF, which shows a strong, dense band extending from the Sierra Nevadas
through the Canadian Rockies. The BELD ENF broadly covers the Rocky Mountain Range, yet more diffusely (land
cover percentage up to 76%), whereas the IGBP dataset shows more sparse and dense ENF land cover (e.g., up to
100%) in the same range. For the DBF category, another notable difference is that the strong band of oaks around
the Central Valley of California, which is evident in BELD but missing from the IGBP data set. Additionally, the
IGBP has far greater densities of DBF along the Appalachian range than BELD. Overall, the IGBP land cover
fractions for forest PFTs are higher on average than the summed BELD taxa.





Grass spatial distributions are given by $C_3$ (non-arctic) and $C_4$ grass PFT land cover classes from IGBP (Figure 4e,f),
which correspond to the observed family-level Poaceae pollen subdivided into $C_3$ and $C_4$ categories (described in
Section 2.2). $C_3$ coverage is evident across the United States, with broad coverage throughout the Southeast,
Midwest, and northern Great Plains (Fig. 4e). $C_4$ coverage is concentrated in the Southeast and Southern Great
Plains at lower densities (Fig. 4f).
Ragweed requires a different land cover treatment as land cover distributions are not available for ragweed across
the entire continental United States. Ragweed is known to arise in areas of human disturbances [*Forman and*
*Alexander*, 1998; *Larson*, 2003], and is found mainly in disturbed or developed areas such as cities and farms [*Clay*
*et al.*, 2006; *Katz et al.*, 2014]. *Ambrosia* land cover (Figure 4g) is derived from the urban and crop categories of the
CLM4 land cover, which are sourced from LandScan 2004 [*Jackson et al.*, 2010] and the IGBP datasets,
respectively. The urban data is subdivided by urban intensity, which is determined by population density. We
assume that ragweed is unlikely to grow in the densest of urban areas (such as city centers), and utilize the lowest
urban density category that is also the most widespread. Ragweed land cover (plants $m^{-2}$) in urban areas is
determined by multiplying the average urban ragweed stemdensity given by *Katz et al.* [2014] by the urban land
cover fraction. For crops, the IGBP subdivides land cover fraction into categories including corn and soybean crops,
and *Clay et al.* [2006] provide ragweed stem densities in soybean and corn cropland. Thus, we calculate the ragweed
land cover fraction ($LC_{rag}$):

(1)    $$f_{rag} = \alpha\left(d_{soy}f_{soy} + d_{corn}f_{corn}\right) + \beta\left(d_{urb}f_{urb}\right)$$

where $d_{soy}$, $d_{corn}$ and $d_{urb}$ represent the stem density of ragweed in soybean, corn and urban areas, respectively, and
the $f_{soy}$, $f_{corn}$ and $f_{urb}$ represent the fractional land cover for soybean, corn and urban, respectively. $\alpha$ and $\beta$ are tuning
parameters to that are determined by a preliminary evaluation between modeled and observed ragweed pollen
counts, where $\alpha = 0.01$ for crop and $\beta = 0.1$. *Zink et al.* [2017] show that a ragweed land cover representation
developed by combining land use and local pollen count information evaluates better against observed pollen counts
than even ragweed ecological models, giving confidence to this choice of land cover representation.
All land cover data are regridded to a 25 km resolution across the United States to provide emissions at the same
spatial resolution as the regional climate model (see Section 5).

**3.2 Meteorological data for phenology**

To develop the emissions model, we use two sources of meteorological data.  The first is a high-resolution
meteorological dataset to develop the phenological relationships for the timing of pollen release. Because reliable
measurements are not available at all pollen count stations and there is uncertainty in the siting of these stations
(e.g., they may be in urban areas with highly heterogeneous temperature), we use a gridded observational
meteorological product for consistency across all sites [*Maurer et al.*, 2002].  The gridded Maurer dataset
interpolates station data to a 1/8º grid across the continental United States on a daily basis, representing a high
spatial resolution gridded data product where data from each met station has been subject to consistent quality
control. Higher resolution DayMet temperatures (daily 1 km) [*Thornton et al.*, 2014] were used in lieu of Maurer





data at NAB sites where the Maurer dataset did not provide information at the collocated grid cell (Table S1). For
offline emission calculations input into the regional climate model, we use annual-average temperatures computed
from monthly Climate Research Unit (CRU) temperature data [*Harris et al.*, 2014]. This data was interpolated from
a 0.5°x0.5° grid to the 25 km regional climate model grid used for pollen transport.

## 4 PECM model description


### 4.1 Emission potential


The pollen emissions model is a prognostic description of the potential emissions flux of pollen ($E_{pot}$; grains m$^{-2}$ d$^{-1}$)
for an individual taxon *i*:

$$(2) \quad E_{pot,i}(x,y,t) = f_i(x,y) \frac{p_{annual,i}}{\int_0^{365} \gamma_{phen,i}(x,y,t)\, dt} \gamma_{phen,i}(x,y,t)$$

for a model grid cell of location *x* and *y* at time *t*. In this expression, *f(x,y)* is the vegetation land cover fraction
(Section 2.1; m$^2$ vegetated m$^{-2}$ total area), $p_{annual}$ is the daily production factor (grains m$^{-2}$ yr$^{-1}$)*,* and $\gamma_{phen}$ is the
phenological evolution of pollen emissions that controls the release of pollen (description below). Equation 2 can
apply to either a single taxa or PFT, depending on the prescription of land cover through *f(x,y)*. In the simulations
described here, emissions are calculated offline based on this equation and provided as input to a regional climate
model (RCM). This emission potential is later adjusted based on meteorological factors in the RCM where the
pollen grains are transported as aerosol tracers (Section 5.1.1). In the future, Equation 2 can be coupled directly
within the climate model for online calculation of emissions. The phenological and production factors are described
in greater detail below.

### 4.2 Phenological factor ($\gamma_{phen}$)


Based on the observed pollen counts, a Gaussian distribution is used to model the phenological timing of pollen
release ($\gamma_{phen}$):

$$(3) \quad \gamma_{phen,i}(x,y,t) = e^{-\frac{(t-\mu(x,y))^2}{2\sigma(x,y)^2}}$$

where *μ(x,y)* and *σ(x,y)* are the mean and half-width of the Gaussian, respectively, and can be determined based on
the start day-of-year (sDOY) and end day-of-year (eDOY) calculated by an empirical phenological model:

$$(4) \quad \mu(x,y) = \frac{sDOY(x,y) + eDOY(x,y)}{2}$$

$$(5) \quad \sigma(x,y) = \frac{eDOY(x,y) - sDOY(x,y)}{a}$$

The fit parameter, *a*, accounts for the conversion between the empirical phenological dates based on a pollen count
threshold and the equivalent width of the emissions curve. Based on evaluation versus observations, *a* = 3 was
selected for initial offline simulations.
Linear regressions of observed sDOY and eDOY from individual pollen count stations versus temperature are used





to empirically determine sDOY and eDOY that drive $\gamma_{phen}$. An important criteria is the grain count used to determine
the sDOY and eDOY, and we utilize a count threshold adaptable to bimodal emission patterns such as those noted
for *Ulmus* and Poaceae. *Sofiev et al.* [2013] selected dates on which the 5th and 95th percentile of the annual index
(annual sum of pollen counts) were reached, while *Liu et al.* [2016] combined a 5 grains $m^{-3}$ threshold with the
additional condition that 2.5% (97.5%) of the annual sum of pollen was reached before the start (end) date. Here,
we implement a pollen count threshold of 5 grains $m^{-3}$ and found this was sufficient to reproduce the observed
seasonal cycle. To account for smaller signals that may be due to count errors (e.g., an exceedance of the 5 grains $m^{-3}$
threshold but not followed by an increase in emissions), we used a moving window with a threshold of 25 grains
$m^{-3}$ for the sum of pollen counts in the nearest 10 neighboring days; when the sum of the neighbors failed to meet
this threshold, the data point was omitted. In this manner, we calculated the sDOY and eDOY for the full 8-year
time series for each taxon at each station. If more than one start or end date was found in a single year at a single
station for a taxon that was not clearly bimodal, only the first set of dates was retained for the linear regression. For
taxa with an observed bi-modal peak, the second peak was treated as a separate taxon (e.g. early and late *Ulmus*, $C_3$
and $C_4$ Poaceae) with a separate phenology. Once the sDOY and eDOY were determined, outliers in these dates
were determined by bounding the data for each taxon at four times the mean absolute deviation of sDOY and eDOY.
Near surface atmospheric temperature (e.g., 2m height) is an important factor of vegetation phenology. In the
interest of having a regional model of emissions that prognostically calculates the start dates, the previous-year
annual-average temperature (PYAAT) based on near-surface atmospheric temperature from *Maurer et al.* [2002]
and *Thornton et al.* [2014] (Section 2.2) is the explanatory variable in the linear regressions. For example, for a start
date of February 2, 2007, the PYAAT would be the mean temperature for the year 2006. For *Pinus* and
Cupressaceae, PYAAT is calculated differently from July 1, 2005 - June 30, 2006 because emissions of these
families begin in the early winter (December). While emissions in this study are calculated using offline
meteorological data, this also could be coupled to a dynamic land surface model to predict reasonably accurate
pollen phenological dates.
To exemplify this method, Figure 5 shows the phenological dates and regression lines for the *Betula* (birch) genus,
with all 13 modeled taxon shown in Figures S1 and S2. The sDOY and eDOY of the pollen season show a moderate
and considerable trend with temperature for most taxa and PFTs (Table 1; Figures S1 & S2). The linear regression
models for sDOY explain 41% of the variance on average for DBF taxa, 47% on average for ENF taxa, 48% for $C_3$
Poaceae, and 8% for *Ambrosia* while having a negligible $R^2$ for $C_4$ Poaceae. For eDOY, the linear regression models
explain 21% of the variance on average for DBF taxa, 29% for ENF, 4% for $C_3$ Poaceae, 32% for $C_4$ Poaceae, and
37% for *Ambrosia*. All trends except $C_4$ Poaceae, late elm, and *Ambrosia* are negative, indicating that warmer
previous-year temperatures result in earlier start and end dates. For most tree taxa, the trend of both sDOY and
eDOY are negatively correlated with PYAAT, with a steeper negative slope for sDOY. The correlation for the
duration of the pollen season (eDOY – sDOY) is then positive for all taxa except Cupressaceae. This suggests that
warmer climates have earlier pollen season start and end dates but longer season lengths.
This agrees with the findings of other empirical modeling studies that suggest the pollen season will, on average,
start earlier with a warmer global climate and have a longer duration [*Parry et al.*, 2007]. Though such an outcome



is more intuitive, there is imperfect agreement that earlier start dates and longer seasons will occur unanimously
throughout the United States region [*Yue et al.*, 2015]. It is understood that photoperiod and the dormancy-breaking
process controlled by chilling temperatures play a significant role [*Myking and Heide*, 1995], and it is generally
accepted that a plethora of other factors, such as plant age, mortality, and nutrient availability also affect observed
phenological dates [*Jochner et al.*, 2013]. However, even without these factors, the current phenological model is
applicable to large regions and provides a clear response of plants to inter-annual climate variability as well as long-
term climate changes. For this first assessment of PECM, we assume that the pollen production factor ($p_{annual}$) does
not change with time and that the phenological model described above captures the main features of pollen
emissions.
**4.3 Annual pollen production ($p_{annual}$)**
The annual pollen production factor ($p_{annual}$) is a measure of how much pollen is produced per vegetation biomass
per year based on literature values. *Molina et al.* [1996] report the annual pollen productivity in grains per tree
measured from a number of representative trees from several taxa. The other tree taxa, grasses and ragweed are
reported in either grains per tree or stem, or grains per unit vegetated area [*Hidalgo et al.*, 1999; *Prieto-Baena et al.*,
2003; *Helbig et al.*, 2004; *Fumanal et al.*, 2007; *Jato et al.*, 2007]. All values are converted to an annual production
factor (grains $m^{-2}$ $year^{-1}$) for each modeled taxon (Table 2). After sensitivity experiments running pollen emissions
in RegCM4, we find that the literature value of $p_{annual}$ for Poaceae provides better agreement with observations when
reduced by a factor of 10 for $C_4$ grass, thus we use this value. To obtain the coefficient of daily pollen production
over the duration of the phenological curve, $\gamma_{phen}$, the integral of the daily pollen production is normalized to $p_{annual}$ as
in Equation 2.
**4.4 Offline emissions simulations**
We calculate emissions offline for two versions of PECM that differ in the land cover input data for woody plants.
The first uses the detailed BELD tree database (Figure 3) for tree pollen emissions (hereinafter the "BELD"
simulation), and the second uses globally based PFT data for tree pollen emissions (Figures 4b and 4d) (hereinafter
the "PFT" simulation). For the grass and ragweed taxa, the emissions calculations are identical between the two
simulations as the input land cover is the same for these two categories. While the family and genus level is useful
for the allergen community, the respective taxon land cover databases needed to develop a global, adaptable model
are not always available. While many plant traits are found to vary quite strongly within individual PFTs [*Reichstein
et al.*, 2014], the PFT convention is accepted and remains in use in climate models, particularly because of the lack
of species-level land cover data at large scales. For the PFT version, pollen counts from individual taxa were
summed within each PFT prior to calculation of the phenological regression (Table 1). We exclude the bimodality in
*Ulmus* for the PFT version because it is the only tree taxon that exhibits this behavior, and late *Ulmus* pollen
emissions are relatively small compared to the major DBF season. The production factors for each PFT are
calculated as the unweighted average of the production factors for all the taxa within the PFT (Table 1).





Figures 6-11 show the monthly climatology of the 2003-2010 emission potential calculated by the offline models
described in Section 4.1 ($E_{pot}$; Equation 2). The seasonal cycle can be clearly identified in the emissions potential,
with the onset of pollen emissions beginning in the warmer south and moving northward along the gradient of
annual average temperature. Colder locales such as those at high elevations can interrupt this general trend. Though
pollen seasons generally end later in the colder parts of the domain just as they start later, modeled pollen emission
seasons tend to be shorter at colder locations for most taxa (about 1 day per 1°C, on average). The highest maximum
emissions for DBF occur over the Appalachian range between April and May for both the BELD and PFT versions
(Figures 6 and 7). For ENF, the maximum occurs in April in the American West for the BELD version where
Cupressaceae land cover is dominant, while it is consistent in magnitude between the Southeast and West Coast for
the PFT-based version (Figures 8 and 9). The grass PFT maximum emissions occur in June in the northern Rockies
for $C_3$ and in September in the South-Central Great Plains for $C_4$ (Figure 10). Ragweed pollen emissions reach their
maximum during September throughout the Corn Belt where soybean and corn crops dominate the land surface,
with local maxima apparent in urban centers (Figure 11).

**5 Emissions implementation and evaluation**
**5.1 Emissions implementation in a regional climate model**
To evaluate PECM, emissions calculated offline are included within a regional climate model to compare simulated
atmospheric pollen concentrations with ground-based observations from the NAB pollen network. The two
phenological pollen emissions estimates (BELD and PFT) described above are prescribed as daily emissions, after
which they are scaled by meteorological factors and undergo atmospheric transport. We use the Regional Climate
Model version 4 [*Giorgi et al.*, 2012], which is a limited-area climate model that includes a coupled aerosol tracer
module [*Solmon et al.*, 2006] that readily accommodates pollen tracers. RegCM4 is based on the hydrostatic version
of the Penn State/NCAR mesoscale model MM5 [*Grell et al.*, 1995] and configured for long-term climate
simulations. In our RegCM4 configuration, we use the Community Land Model version 4.5 (CLM4.5; [*Oleson et
al.*, 2010]), the Emanuel cumulus precipitation scheme over land and ocean [*Emanuel*, 1991], and the SUBEX
resolvable scale precipitation [*Pal et al.*, 2000]. Two 8-year simulations of pollen emissions and transport in
RegCM4 were conducted from 2003-2010 with the BELD and PFT version of the offline emissions model. Six
months of spin-up (July-December 2002) are run for both simulations that we exclude from the following analysis.
In the model, we calculate the fate of four pollen tracers corresponding to the four PFTs (DBF, ENF, GRA and
RAG) from the PECM offline emissions. Because individual tracers add to the computational cost of the
simulations, BELD-based tree emissions are summed into DBF and ENF PFTs and emitted into the model
atmosphere. To calculate the emissions, the emission potential calculated offline for each PFT ($E_{pot}$) is scaled
according to surface meteorology following the methods of *Sofiev et al.* [2013]:

(5)    $E_{pollen,i}(x, y, t) = E_{pot,i}(x, y, t)f_w f_r f_h$

(6)    $f_w = 1.5 - e^{-(u_{10} + u_{conv})/5}$







$$(7) \quad f_r = \begin{cases} 1, & pr < pr_{low} \\ \dfrac{pr_{high} - pr}{pr_{high} - pr_{low}}, & pr_{low} < pr < pr_{high} \\ 0, & pr > pr_{high} \end{cases}$$

$$(8) \quad f_h = \begin{cases} 1, & rh < rh_{low} \\ \dfrac{rh_{high} - rh}{rh_{high} - rh_{low}}, & rh_{low} < rh < rh_{high} \\ 0, & rh > rh_{high} \end{cases}$$

where $f_w$, $f_r$, and $f_h$ are the wind, precipitation and humidity factors, respectively. The meteorological parameters in
these equations are from online RegCM variables, including $u_{10}$ and $u_{conv}$ as the 10-meter horizontal wind speed and
vertical wind speed, and $pr$ and $rh$ are precipitation and relative humidity with low and high thresholds. These
scaling factors account for the effects of wind, precipitation and humidity on the emission of pollen from flowers
and cones. The humidity and precipitation factors are piecewise linear functions of the near-surface (10 m) RH and
total precipitation and range from 0 (high precipitation or humidity) to 1 (no precipitation or low humidity). The
wind factor ranges from 0.5 to 1.5, as even in calm conditions turbulent motions can trigger pollen release with high
winds releasing more pollen. These scaled emissions are then transported according to the tracer transport equation
(Equation 9) of *Solmon et al.* [2006] that includes advection, horizontal and vertical diffusion ($F_H$ and $F_V$),
convective transport ($T_c$), as well as wet ($R_{Wls}$ and $R_{Wc}$, representing large scale and convective precipitation
removal) and dry deposition ($D_d$) of an individual tracer ($\chi$), represented by i = 1 to 4 for each PFT pollen emission:

$$(9) \quad \frac{\partial \chi^i}{\partial t} = \bar{V} \cdot \nabla \chi^i + F^i{}_H + F^i{}_V + T^i{}_C + S^i - R^i{}_{Wls} - R^i{}_{Wc} - D^i{}_d$$

### 424     5.1 Model evaluation against observations

We evaluate the efficacy of PECM in simulating the timing and magnitude of pollen emissions across the
continental United States by evaluating RegCM4 tracer concentrations versus observations. We compare the daily
climatology of simulated near-surface pollen counts and observed, ground-based pollen counts for each of the four
modeled PFTs (Figure 12). The observed climatological pollen timeseries are the average of the daily climatology at
all pollen counting stations comprising each of the five major U.S. subregions (Section 2.2) and are compared with
the modeled climatology, which averages the individual grid cells that contain the pollen counting stations.
Interannual variability is assessed using the relative mean absolute deviation for each day of the climatology. The
inter-annual variability in observed daily pollen counts throughout the year is, on average, 81, 78, 78 and 77% of the
mean (DBF, ENF, grass and ragweed, respectively), while this variability from the simulations is 53% for the BELD
version of the DBF model and 61% for the PFT version, 55% and 92% for the BELD and PFT versions of the ENF
model, 43% for grasses, and 49% for ragweed (Figure 12). This indicates that the model is capturing the relative
inter-annual variability of the pollen counts between PFTs, but not all of the variability in pollen counts from season
to season. The unexplained variability in pollen concentrations could be due to the lack of sensitivity of annual
pollen production factor to the environment, as this may be closely tied with precipitation [*Duhl et al.*, 2013] or





temperature [*Jochner et al.*, 2013]. Additionally the climatological pollen count is analyzed using box-and-whisker
plots to assess the models' representivity of pollen count magnitude in spite of phenology (Figure 13). These
metrics are discussed in detail by PFT and U.S. subregion below.
**5.2.1 DBF**
In the Northeast, the BELD model captures both the observed seasonal timing and the magnitude of DBF pollen
counts (Figure 12a). Observed DBF phenology is also simulated by the PFT-based emissions with even greater
statistical accuracy in reproducing the observed pollen counts, though the BELD model more accurately reproduces
the annual maximum (Figure 13a). The accuracy in this subregion is not surprising, as Northeastern pollen counting
stations contributed the greatest number of data points to the phenological regression analyses. Observed DBF
pollen counts in the Southeast have a large maximum that is greater than the climatological seasonal maximum of all
four other subregions and all three other PFTs (Figure 12b), which is predominantly from *Quercus*. Neither the
BELD nor PFT version of the simulation recreates this sharp peak, but they do simulate a large majority of the
pollen count distribution (Figure 13b), especially the PFT-based model for which the lower 75% of simulated
climatological pollen counts agrees well with the lower 75% of observed climatological pollen counts. The PFT
model does not specifically resolve *Quercus*, and while the BELD model does resolve *Quercus*, it fails to model this
maximum. This may be because the linear regression producing the phenological dates is an average, where a
longer season may result from earlier start dates and/or later end dates which will reduce the maxima in the Gaussian
distribution. In the Mountain region, there is an observed maximum early in the spring that is not simulated by either
model because the DBF phenology at several cold Mountain sites is exceptionally early, and falls well below the
regression lines (Figures S1, S2). However, both the BELD and PFT model simulate the second Mountain subregion
peak with the correct magnitude. The BELD simulated maximum DBF in California is about 40 days later than the
observed peak, also due to the regionally anomalous phenology in California as compared with the rest of the U.S.,
and though the PFT model peaks much closer to the observations, it underestimates DBF pollen counts. In the
Pacific Northwest, the observed pattern is quite similar to the DBF pollen phenology in the Mountain subregion with
only a slightly weaker early spring peak due to low-elevation pollen. The observed phenological pattern (Fig. 12e)
and pollen count magnitudes (Fig. 12e) are both more accurately simulated by the BELD model, likely due to the
earlier spring maximum that does not appear in the PFT simulation.
**5.2.2 ENF**
Like DBF, the BELD ENF in the Northeast is well represented by simulating two distinct Cupressaceae and
Pinaceae maxima, although the model slightly underestimates observed Pinaceae pollen counts (Figure 12f). The
PFT model ENF phenology emits from the start of the earlier Cupressaceae season to the end of the later Pinaceae
season, while overestimating the maximum pollen count by about a factor of 2. In the Southeast, the winter peak is
not captured by the model (Figure 12g) due to negligible BELD land cover fractions for Cupressaceae (Figure 4c).
However, the spring Pinaceae maximum is accurately captured by the BELD simulation. The PFT model follows the
observed Pinaceae phenology more closely, though overestimating pollen counts by a factor of 2 to 3 and estimating



a later ending date by about 40 days. In the Mountain subregion, ENF start and end dates are simulated by the BELD
model with improved accuracy than the DBF phenology in this subregion, though the predicted spring maximum is
later than observed (Figure 12h). As with DBF, there is good agreement between the BELD model with the later part
of the season in this subregion. The PFT model, again, simulates the peak ENF emissions in the later part of the
season and overpredicts the pollen counts by a factor of 2 to 3. In the California subregion, the tails of the pollen
distributions by both models closely resemble the pollen count magnitudes, yet the majority of these pollen counts
(the top 75%, Figure 13i) lie above the observed maximum (Figure 12i). Finally, in the Pacific Northwest, the
BELD model phenology shows some agreement with the model mean (Figure 13j), with the simulated pollen count
showing a stronger Gaussian distribution than observed (Figure 12j).  In contrast, the PFT model grossly
overpredicts the observed pollen counts by up to a factor of 10 at its maximum, likely due to the greater
representation of the ENF PFT than the BELD model in this region. The simulated climatological start date of the
PFT model is within a few days of the observed climatological start date, while the end date is about 20 days later
than observed.
**5.2.3 Grasses**
Grass phenology across all subregions for both $C_3$ and $C_4$ types is captured by the emissions estimates (Figure 12 k-
o). However, the pollen count magnitude in Northeastern $C_3$ grass peak is overestimated by about a factor of seven,
even when using the minimum value of the annual production factor in the range estimated by *Prieto-Baena* (2003)
(Figure 12k).  The secondary peak, which we attribute to $C_4$ grasses and is only about half as large, is well-
represented. In the Southeast, the simulated pollen count magnitudes are much closer to observations, while the $C_3$
peak is overestimated here by only a factor of 2 and the $C_4$ peak is within 5 grains m$^{-3}$ (Figure 12l).   In this region,
the observed duration of the pollen emissions is not fully captured by the simulated grass phenology in the
Southeast, and this is probably due to the non-Gaussian shape of the observed time series. In the Mountain
subregion, the $C_3$ pollen count is overestimated by the model, but the phenology is represented by a gradual rise in
low emissions beginning in March to match the maximum burst of emissions in June (Figure 12m). $C_4$ grass pollen
counts are not simulated in the Mountain region due to the relatively low $C_4$ land cover in the IGBP dataset (Figure
4f). In California there is a single observed grass peak, which the model attributes to $C_3$ pollen, and the peak count
in the simulation is about 5 days late and about 2 to 10 times too large (Figure 12n). In the Pacific Northwest, the
climatological $C_3$ season is accurately simulated with the exception that the phenology is shifted 20 days earlier than
observed  (Figure 12o). A small $C_4$ peak in the observations at around day 260 is not simulated in this region due to
negligible land cover for $C_4$ grasses in the IGBP land cover data (Figure 4f).
**5.2.4 Ragweed**
Simulated ragweed phenology in the Northeast, Southeast, and Mountain subregions follows the observed
phenology very closely, where the peaks of both the simulated and the observed climatology occur within a day of
each other (Figures 12p-t). Close evaluation of each regional phenological timeseries reveals that many of the
observed features, like those determined by the rate of increase or decrease of the pollen count, are reproduced by



the model. The magnitude of the modeled ragweed maxima in the Northeast and Mountain subregions is slightly
greater than observed (Figures 12p and 12r), while there is a clear underestimation by a factor of 4 or 5 in the
Southeast (Figure 12q). The observed climatological ragweed pollen counts in California and the Pacific Northwest
are negligible, though the simulation predicts them to be similar in magnitude and timing to the other three
subregions (Figure 12s and 12t). These discrepencies may be due to the land use description developed for ragweed
(Section 3.1), which may overestimate the ragweed potential in the western United States, or potentially the
relatively spare observational stations in these regions may be poorly placed relative to emissions sources.

## 6 Conclusions

We have developed a climate-flexible pollen emissions model (PECM) for the 13 most prevalent wind-pollinating
taxa in the United States based on observed pollen counts. PECM was adapted to the PFT categorization common to
climate and Earth system models with four major temperate-zone PFTs (DBF, ENF, grasses and ragweed), thus it is
possible to apply this model to larger geographic regions where specific taxon-level data is unavailable. We
evaluated PECM using a regional climate model (RegCM4) to transport emissions and evaluated resulting pollen
counts versus observations. PECM generally captures the observed phenology, and observed surface pollen
concentrations can be simulated within an order of magnitude. While many emissions models to date have focused
on smaller geographical regions with more detailed land cover information and pollen information, this model
represents the first of its kind to simulate multiple taxa over broad spatial areas. This transition to a larger scale does
have its disadvantages, and we define several major sources of uncertainty to consider when scaling up pollen
emissions to the regional or global scale: (1) pollen production factors, (2) climatic sensitivities in phenological
timing, (3) land cover data, and (4) taxa specificity. We discuss each of these uncertainties in greater detail.
A large source of uncertainty is the use of a constant annual production factor for pollen (Section 4.3). It has been
reported that wind-driven pollen production has increased historically and is expected, potentially, to increase in the
near future [*Parry et al.*, 2007; *Zhang et al.*, 2014]. Some of more effective improvements to the emission model
would be to create a pollen production model that is sensitive to multiple environmental factors such as soil
moisture, temperature and nutrient status [*Jochner et al.*, 2013]. The interannual variability in observed daily pollen
counts is, on average, substantially greater than that of the modeled pollen counts, which is likely due to this lack of
production sensitivity. The current production factors for woody plants could be enhanced by studies that extend the
number of representative units (i.e. individual trees) of vegetation used to determine the average pollen production.
In a PFT representation, there is an inevitable limitation to the accuracy of any single PFT's ability to account for
taxa differences within the PFT. Furthermore, the current model also assumes that there are no interspecies
differences that affect the performance of the BELD model as well as the PFT model, whereas in reality it may vary
by an order of magnitude within a genus [*Duhl et al.*, 2013]. However, despite the assumption of a constant
production factor, observed surface pollen counts for all PFTs are typically reproduced within a single order of
magnitude, as apparent in emission model evaluation.



Second, the use of observed relationships between pollen count and temperature to determine the phenological
pollen start and end date also adds uncertainty to our modeling framework. Firstly, we assume stationarity in the
phenological relationships, and this assumption may be violated. Secondly, based on the subregions defined for the
analysis, there appears to be a bias in the linear regressions toward subregions with more available pollen counting
stations, therefore affecting performance differences in these regions. Lastly, even though generally the Gaussian
time series model of the pollen phenology performs well in our analysis, in the PFT representation the Gaussian
absorbs or misses some of the phenological details in the observed pollen seasonality, and in some cases taxa (e.g.
grasses in the Southeast subregion) may not be captured by the existing phenology.
Third, the specificity of land cover data provides an important constraint in the overall simulation of emissions. The
representation of land cover is a key factor to accurately capturing regional features, especially in areas with a high
degree of topographical variation and therefore greater variance in the land cover.  For example, we notice large
differences in the two model simulations when considering tree-specific taxa, such as in the western United States
for ENF (Section 5.2.2).  Also, our definition of the land cover available for ragweed used assumptions based on
crop cover and urban area, which overestimated emissions in the western United States (Section 5.2.4).
Interestingly, even though ragweed lacks an exact spatial distribution, distinct observed features of the ragweed
phenology in three of the five subregions emerged using the current ragweed land cover parameterization.
Fourth, the aggregation of emissions to the PFT level affects the representativeness of the production factors,
phenology and land cover.  When comparing the two models of the tree pollen (BELD versus PFT), the individual
phenology of each of the 11 tree taxa are resolved by the BELD simulation, whereas they are either folded into or
excluded from the single phenology modeled by the PFT simulation. This results from either treating the taxa in the
phenological regressions individually, as in the BELD model, or as a sum, as in the PFT model. With a few
exceptions (e.g., the ENF family distinctions), the PFT model does generally reproduce the regional phenology
throughout the United States domain, which is a priority of this study.
Despite these limitations, the empirical formulation presented here is the first of its kind to predict a broad range of
different pollen emissions across a large geographic region. Even with univariate phenology and invariable pollen
production factors, the model includes seasonal dynamics sensitive to climate change consistent with observations
and is also able to simulate observed pollen magnitudes.  As a result, the model can be useful for estimating of how
allergenic risk or plant reproductive potential will be redistributed by climate change, as well as studying pollen as
an aerosol in the climate system. While the empirical phenological models can be reproduced for any set of regional
pollen counting stations, PECM as a whole can be easily adapted to various community climate and earth system
models, global and regional, to extend research on the relationships and interactions between pollen and climate.

**7 Code and Data Availability**
Source code for Pollen Emissions for Climate Models (PECM) is written as FORTRAN90 (*.f90) and available in
the supplementary material as plain text. Input data is explained in Section 3 of this manuscript.



**8 Acknowledgements**
This work was supported by NSF Grant to AGS 0952659 to ALS. We thank Melissa Zagorski and Yang Li of
University of Michigan for contributions to the emissions model development, and Fabien Solmon and Li Liu of the
International Centre for Theoretical Physics for RegCM support and their prior work. We gratefully acknowledge
the use of the American Association for Allergy Asthma and Immunology (AAAAI) pollen count data, with
individual station acknowledgments fully provided in Table S1.





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





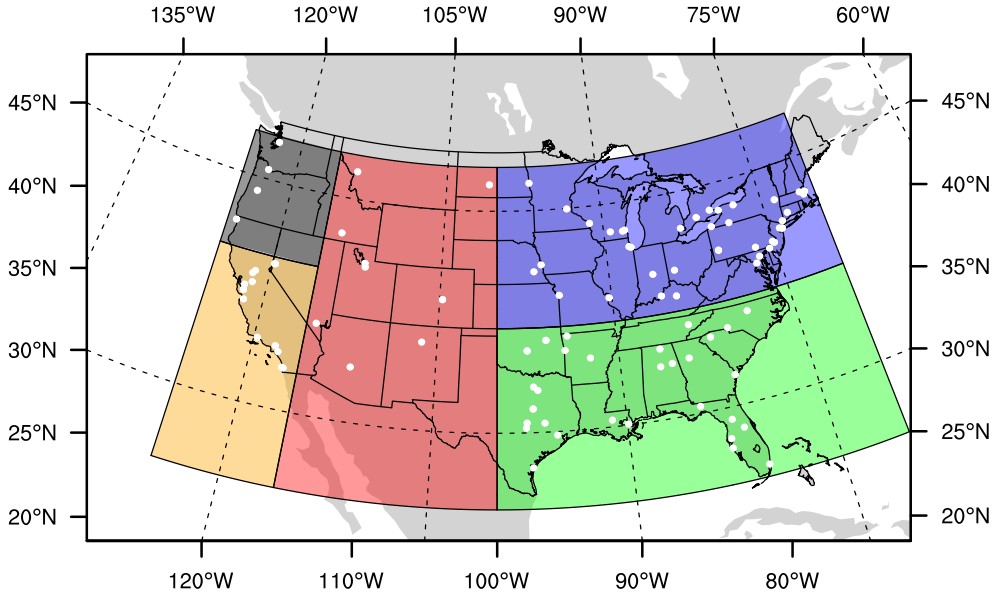


**Figure 1.** **Locations of AAAAI station locations and geographic regions used in this study: Northeast (NE; >38°N, <100°W) in blue, Southeast (SE; <38°N, <100°W) in green, Mountain (MT; 100°W to 116°W) in red, California (CA; <40°N, >116 °W) in orange, and Pacific Northwest (PNW; >40°N, >116 °W) in black.**




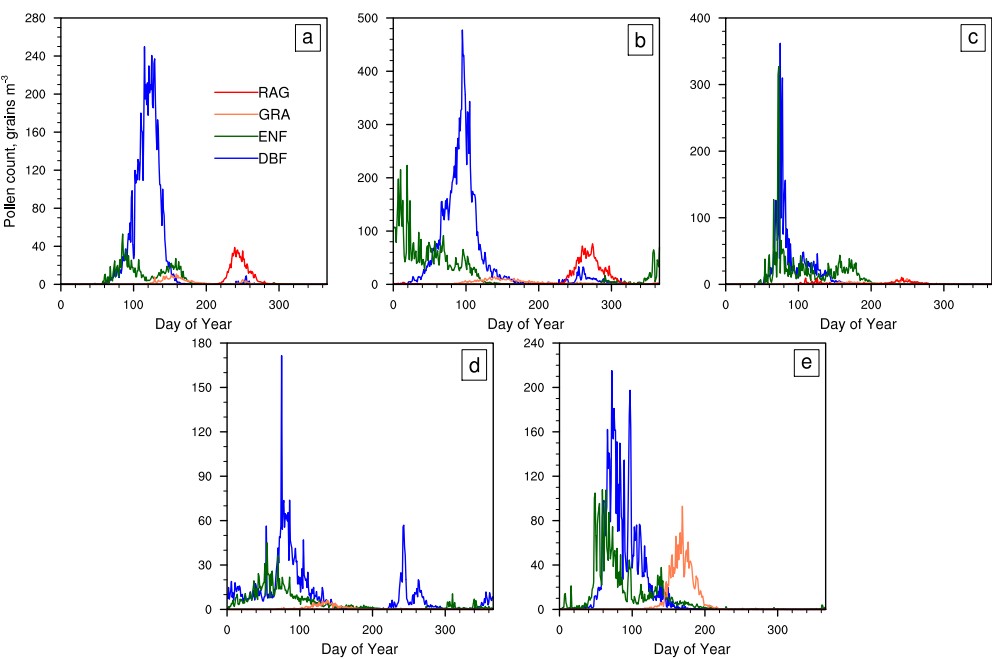

**Figure 2.** **Daily observed climatological time series of pollen count data (2003-2010) for the four representative plant functional types (DBF, ENF, grasses, ragweed) averaged over the five regions in Figure 1: (a) Northeast, (b) Southeast, (c) Mountain, (d) California, and (e) Pacific Northwest.**



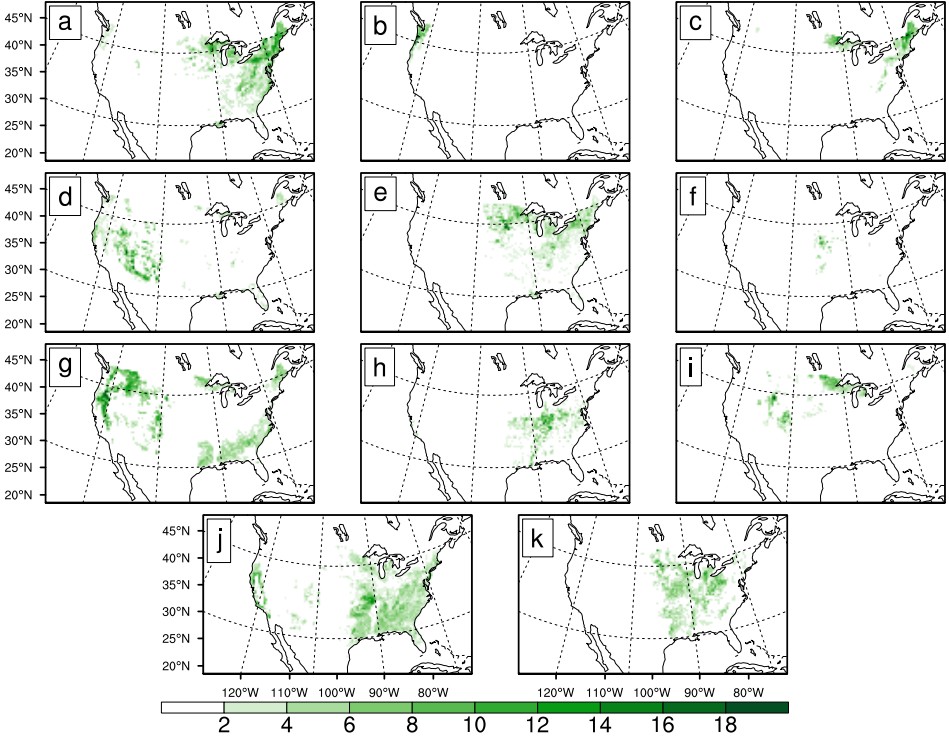


**Figure 3.  Land cover fraction (% coverage) for 11 tree taxa from the Biogenic Emissions Land cover**


**Database (BELD3) regridded to a 25km resolution grid, including:  a)** *Acer* **(maple), b)** *Alnus* **(alder), c)**


*Betula* **(birch), d) Cupressaceae (cedar/juniper), e)** *Fraxinus* **(ash), f)** *Morus* **(mulberry), g) Pinaceae (pine), h)**


*Platanus* **(sycamore), i)** *Populus* **(poplar/aspen), j)** *Quercus* **(oak), k)** *Ulmus* **(elm).**





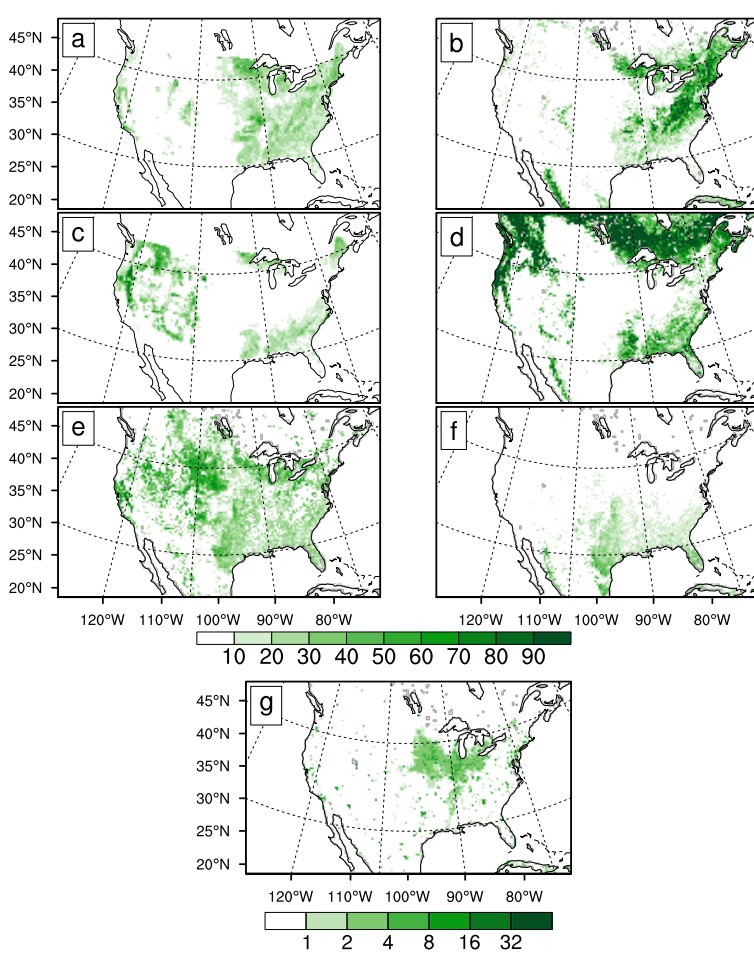


**Figure 4. BELD3 (a, c) and IGBP (b, d, e, f) land cover for the four PFT categories that produce pollen emissions, including (1) deciduous broadleaf forest for (a) BELD3 and (b) IGBP, (2) evergreen needleaf forest for (c) BELD3 and (d) IGBP, (3) grasses, including (e) C3 grasses and (f) C4 grasses, and (4) ragweed, represented by crop and urban IGBP categories.**






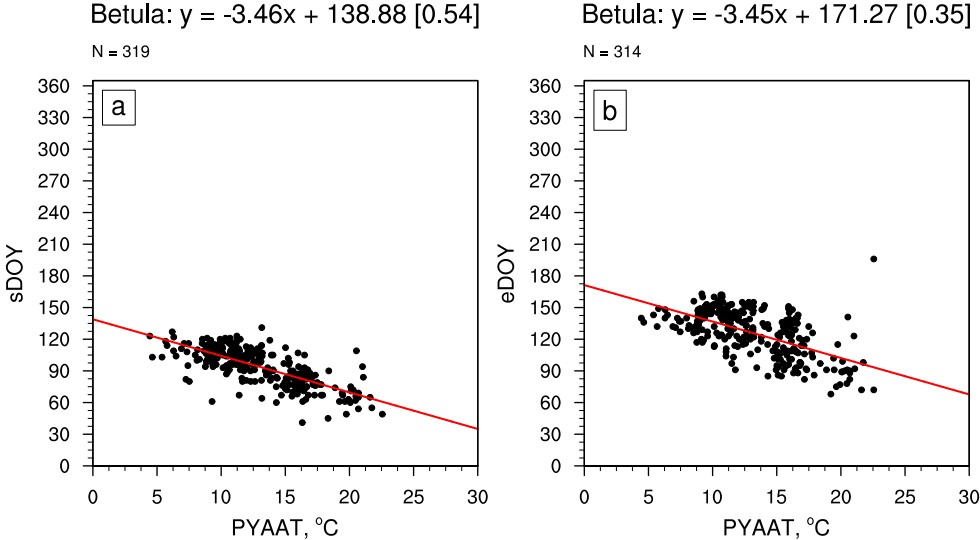


**Figure 5. Phenological regressions for *Betula* (birch) pollen for (a) Start Day of Year (sDOY) and (b) End Day**

**of Year (eDOY) versus previous year annual average temperature (PYAAT; ºC). Each point signifies one**

**station per year for pollen count data from 2003-2010 (total denoted as N).**




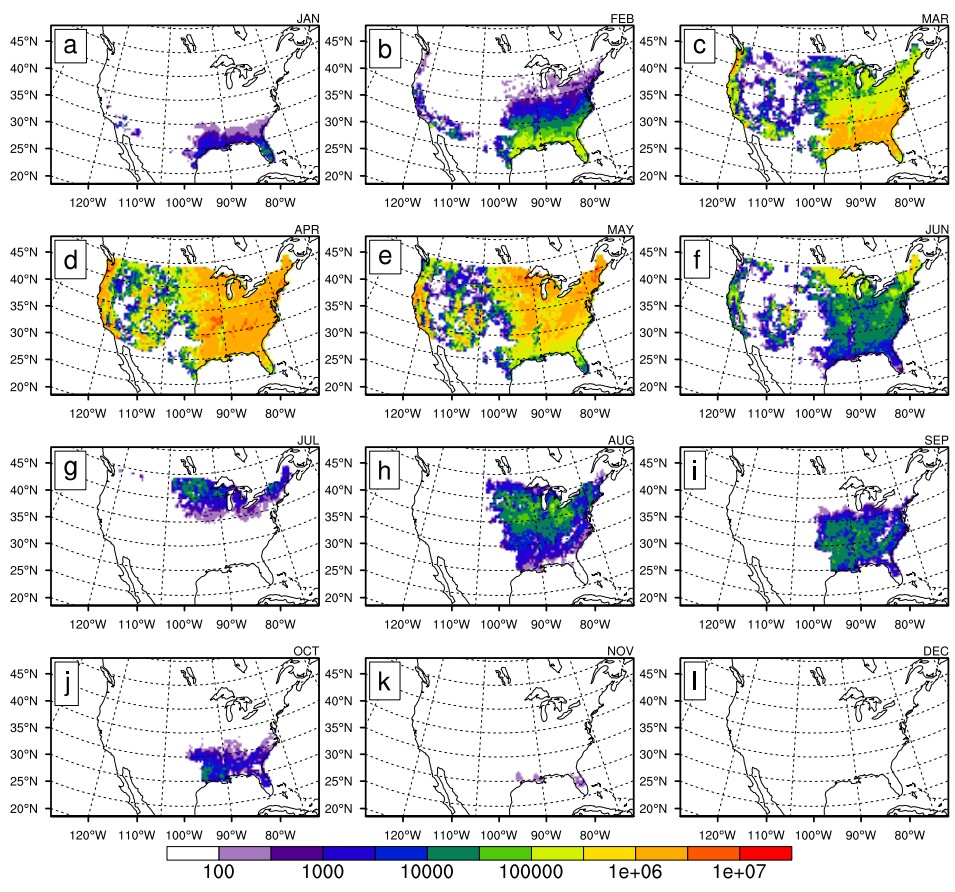


**Figure 6. Monthly climatological emissions potential (E; Equation 1) for BELD model DBF (2003-2010), in**

**grains m$^{-2}$ day$^{-1}$. a) January, b) February, c) March, d) April, e) May, f) June, g) July, h) August, i)**

**September, j) October, k) November, l) December.**




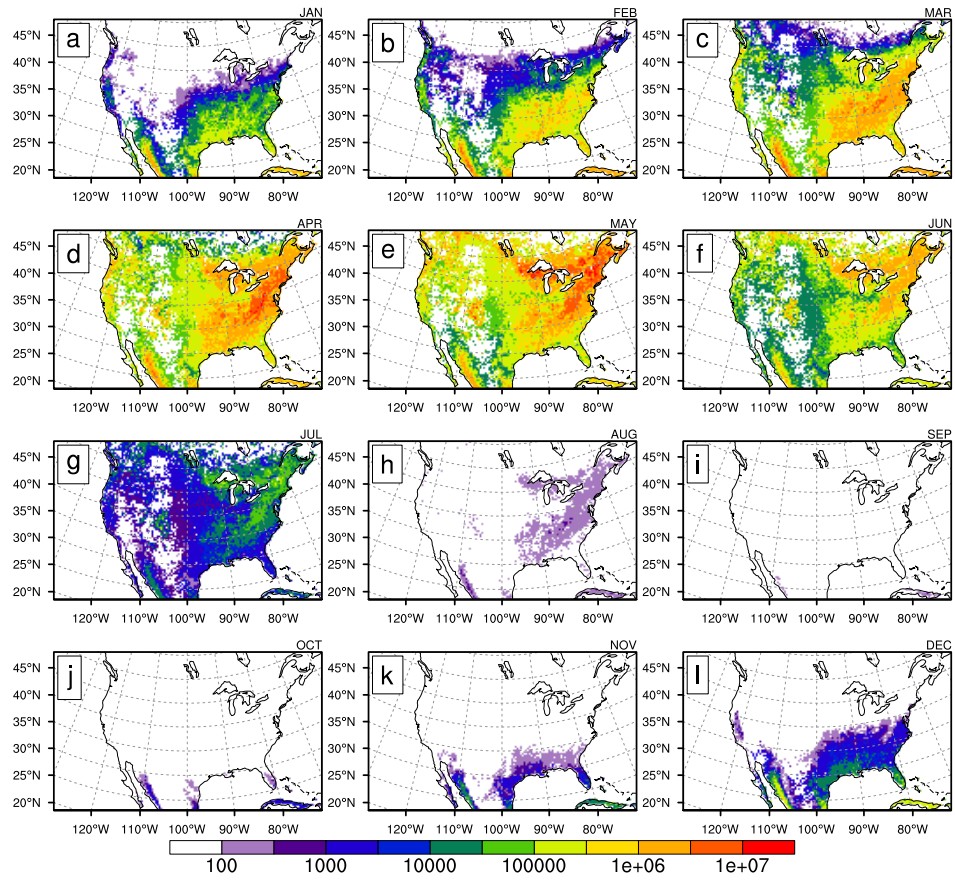


**Figure 7. Same as Figure 6, but for PFT model DBF.**




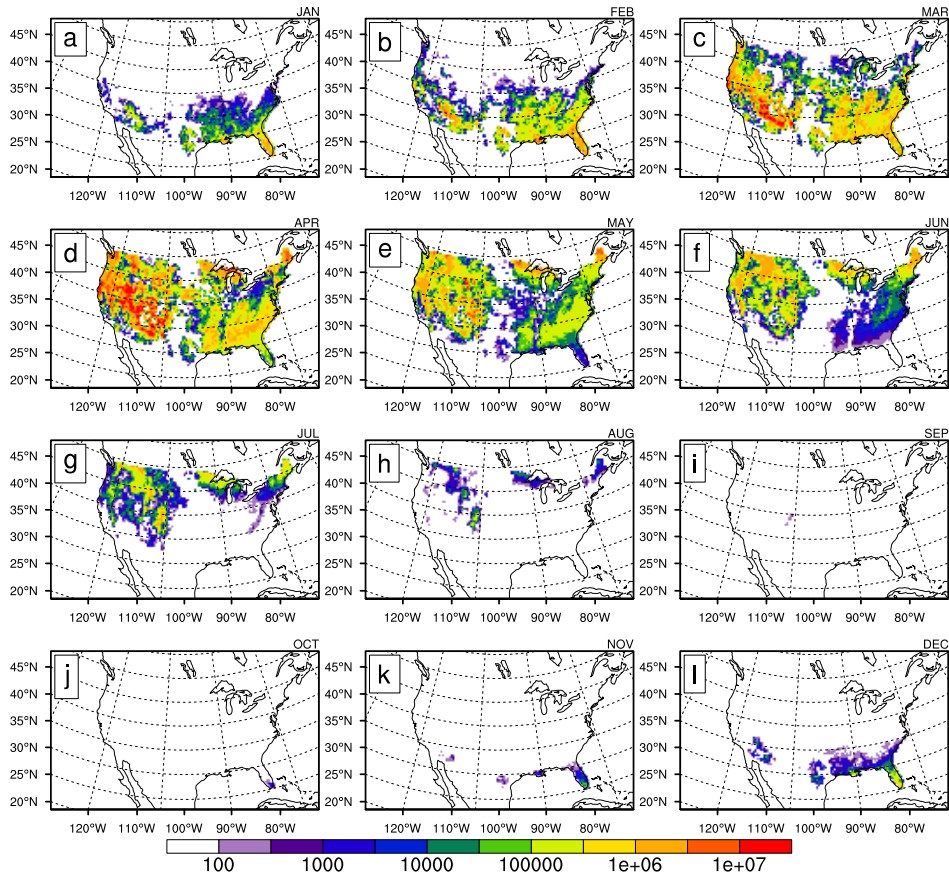


**Figure 8. Same as Figure 6, but for BELD model ENF.**




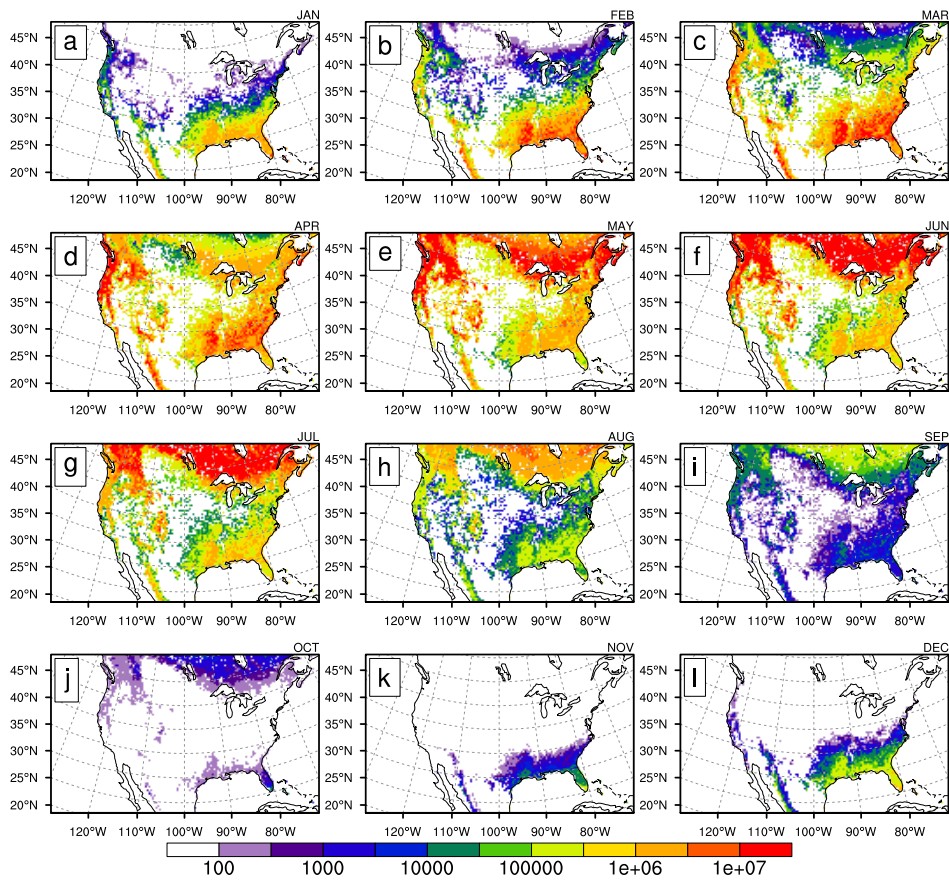

**Figure 9. Same as Figure 6, but for PFT model ENF.**



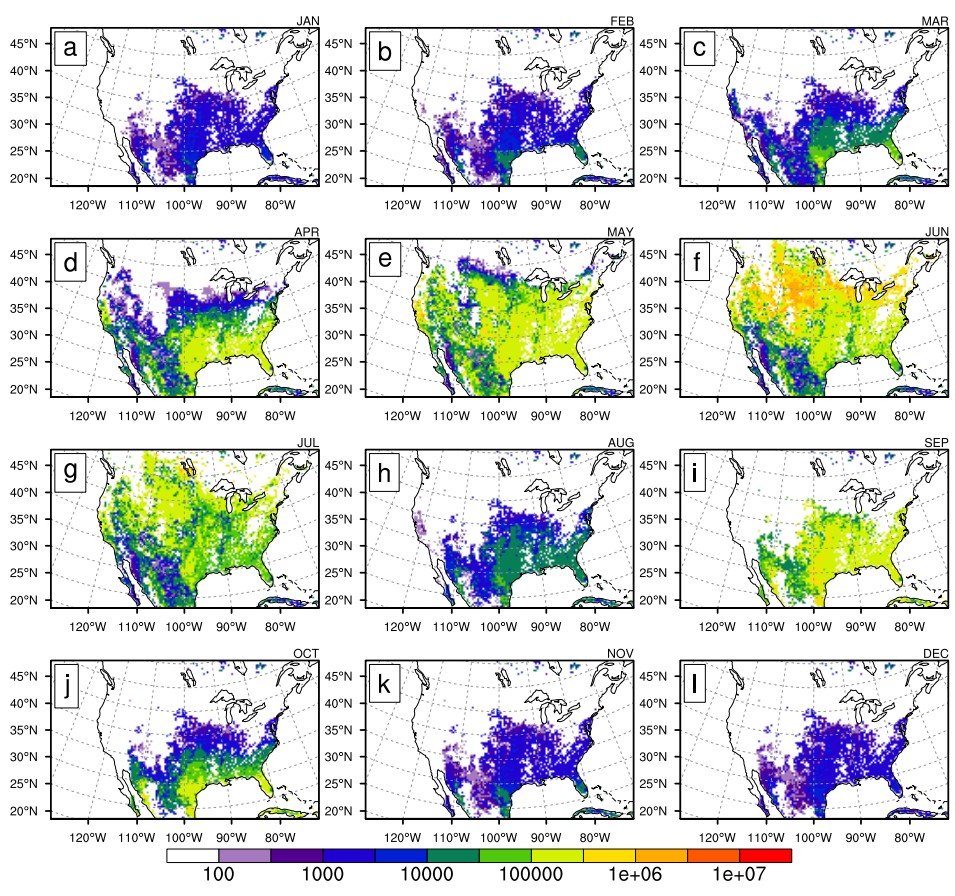


**Figure 10. Same as Figure 6, but for C₃ + C₄ grass.**






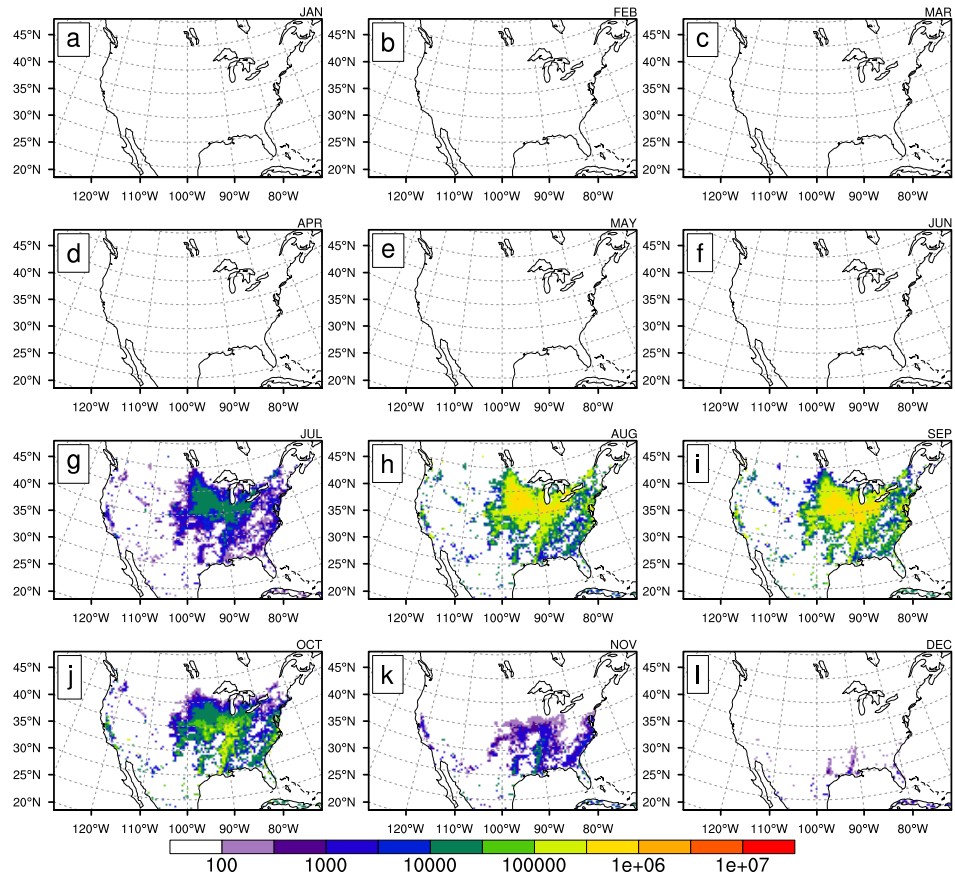


**Figure 11. Same as Figure 6, but for ragweed.**


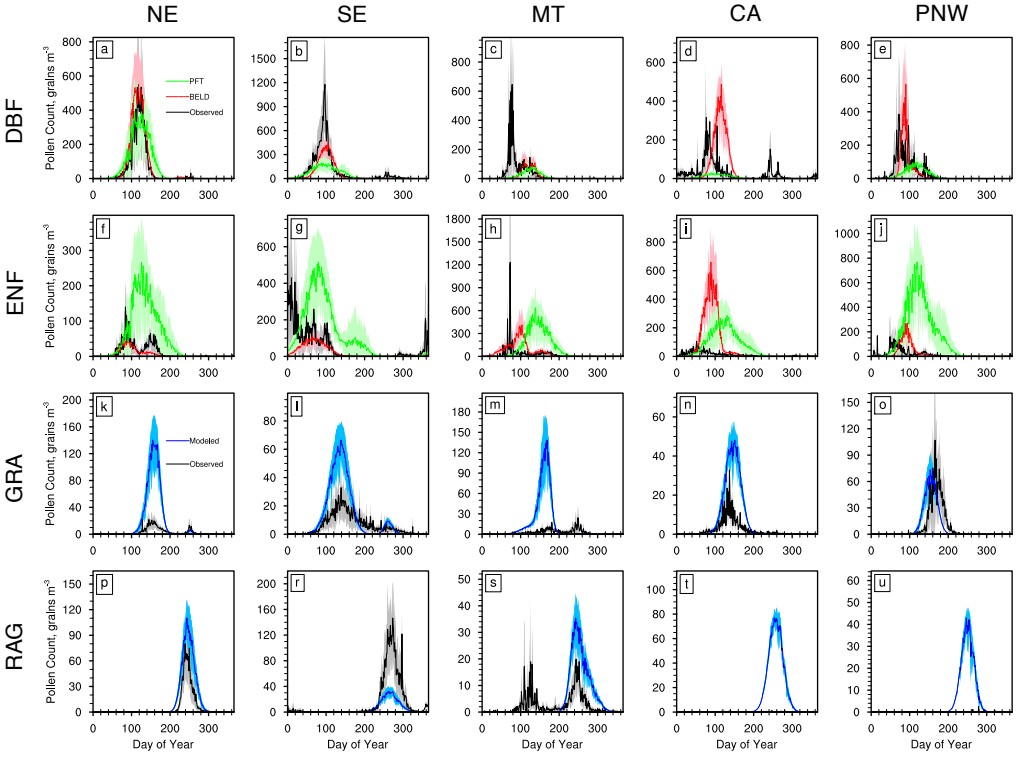


**Figure 12.  Daily climatological (2003-2010) time series of pollen counts comparing model and observations for four PFTs (a-e, deciduous broadleaf, DBF; f-j, evergreen needleleaf, ENF; k-o, grasses, GRA; p-u, ragweed, RAG) across 5 U.S. subregions (columns from left to right: Northeast, NE; Southeast, SE; Mountain, MT; California; Pacific Northwest, PNW).   Shading for the observations and model represents the mean absolute deviation from the climatological average for each day of the climatology. Note: scale of y-axes varies.**






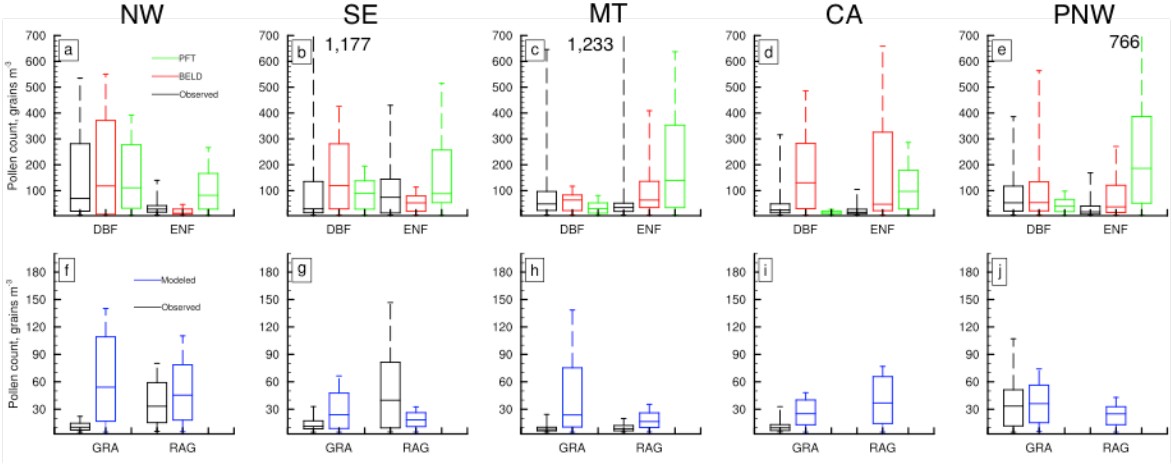

**Figure 13. Box-and-whisker plots showing the statistical spread of the pollen count magnitudes from the climatological curves from Figure 12. Box and whiskers from bottom to top represent the minimum, lower quartile, median, upper quartile, and maximum. Maxima that are not visible in panels b, c and e are noted on plot as a number. All y-axes are same scale.**




| TAXON | P (10^7 grains m^{-2}) | sDOY (slope/R^2) | eDOY (slope/R^2) |
|---|---|---|---|
| **Deciduous Broadleaf Forest (DBF)** | | | |
| *Acer* | 89.1 | -1.78/0.15 | -1.56/0.06 |
| *Alnus* | 210 | -8.82/0.46 | -4.88/0.26 |
| *Betula* | 140 | -3.46/0.54 | -3.45/0.35 |
| *Fraxinus* | 45.1 | -4.69/0.50 | -2.92/0.32 |
| *Morus* | 10 | -4.00/0.53 | -2.97/0.29 |
| *Platanus* | 121 | -4.47/0.40 | -2.65/0.2 |
| *Populus* | 24.2 | -2.23/0.24 | -0.31/<0.01 |
| *Quercus* | 78 | -4.09/0.53 | -2.03/0.19 |
| *Ulmus (early,late)* | 3.55 | -4.61/0.59, 3.06/0.12 | -2.37/0.16, 5.12/0.29 |
| **DBF** | 80.1 | -4.55/0.46 | -1.94/0.13 |
| **Evergreen Needleleaf Forest (ENF)** | | | |
| Cupressaceae | 363 | -5.67/0.48 | -2.67/0.17 |
| Pinaceae | 22.2 | -5.72/0.45 | -5.03/0.41 |
| **ENF** | 193 | -5.95/0.4 | -4.96/0.33 |
| **Grasses (GRA)** | | | |
| **Poaceae (C₃,C₄)** | 8.5, 0.85 | -4.76/0.48, 0.05/<0.01 | -1.08/0.04, 2.96/0.32 |
| **Ragweed (RAG)** | | | |
| *Ambrosia*[a] | 119 | 1.08/0.08 | 3.42/0.37 |

[a] *Ambrosia* production factor in $10^7$ grains plant$^{-1}$.

**Table 1: Production factors (P) and phenological regression coefficients for the start day of year (sDOY) and**
**end day of year (eDOY) as a function of temperature for the 13 individual pollen-producing taxa. Individual**
**taxa and families are organized into the four PFTs, with the two aggregated tree PFTs in gray shading.**
**Regression slope (days/ºC) and coefficient of determination are provided for both sDOY and eDOY**
**(slope/R$^2$).**