# Peer review of "A prognostic pollen emissions model for climate models (PECM1.0)"

_Geoscientific Model Development, 2017_

## Referee Comment (RC1) · Anonymous Referee #1 · 7 Jul 2017

This is an outstanding body of research for which I congratulate the two authors (Matthew Wozniak and Allison Steiner). The development of models of pollen emissions for use within regional and global climate models to simulate pollen counts over the seasonal cycle based on geography, vegetation type and meteorological parameters is cutting-edge research with, as the authors point out, a number of important applications.

The manuscript is beautifully written and presented.

Thank you.

The following comments and corrections will further improve this already excellent manuscript.

[Figure]

General: Throughout the manuscript the word "climatological" or "climatology" is used instead of the word "average" or "mean" with regard to pollen. For example, lines 161-162 state that: "For deciduous broadleaf forest (DBF) taxa, the Southeast has the highest climatological pollen maximum reaching up to about 700-1200 grains m-3 around day 100.". This is confusing because it is applied to a non climatic/meteorological variable, and because it is used in a manuscript which also focusses on climate. It would be much better to simply state: "For deciduous broadleaf forest (DBF) taxa, the Southeast has the highest average pollen maximum reaching up to about 700-1200 grains m-3 around day 100.".

Similarly, lines 159-160 could be modified from: "Figure 2 shows the observed climatological PFT daily pollen counts averaged over all stations within the defined subregions." to: "Figure 2 shows the observed average daily PFT pollen counts averaged over all stations within the defined subregions.".

And so on.

Line 17, Abstract: "PFT" is used without being given in full earlier in the Abstract, so please provide both the full and abbreviated form here.

Line 40, Introduction: The authors may wish to refer to two recently published works that relate to the introductory material here and/or elsewhere in the Introduction:

Sofiev M, Prank M. Impacts of climate change on aeroallergen dispersion, transport, and deposition. In: Beggs PJ (Editor). Impacts of Climate Change on Allergens and Allergic Diseases. Cambridge University Press, Cambridge, 2016. pp 50-73.

Beggs PJ, Šikoparija B, Smith M. Aerobiology in the International Journal of Biometeorology, 1957–2017. International Journal of Biometeorology 2017. DOI: 10.1007/s00484-017-1374-5 [see section on "Aerobiological modelling and forecasting"]

Line 98: This sentence makes reference to "the Finnish emergency modeling system

(SILAM)". It would be better to change this to "the Finnish System for Integrated mod-eLling of Atmospheric coMposition (SILAM)" as given in the Introduction section of Sofiev et al. (2013).

Line 130, paragraph 1 of section 2.1: Table 1 does not relate to "NAB pollen count data ranging from 2003-2010 at all stations in the continental United States", so delete reference to it and just refer to Figure 1.

Line 139: This line includes a reference to Table 2. There is no Table 2 in the manuscript. Should it be Table S2?

Line 140: Change "Cupresseceae" to "Cupressaceae".

Lines 155-159, paragraph 1 of section 2.2: Currently just two of the four boundaries of each of the five subregions are provided. Please provide upper and lower limits of both latitude and longitude for each subregion.

Lines 161-170: The paragraph in these lines seems to contain several values that do not match what is shown in Figure 2. Specifically, deciduous broadleaf forest (DBF) taxa in the Southeast does not have an average pollen maximum reaching up to about 700-1200 grains m-3. Figure 2b shows that it only reaches up to about 500 grains m-3. In the Northeast, DBF does not reach up to an average of 400 grains m-3. It peaks just above 240 grains m-3. And finally, a sharp maximum of 775 grains m-3 does not appear in the Mountain subregion. The sharp maximum is only about 360 grains m-3. The paragraph should be carefully checked.

Line 173: As above, please check the numbers 400 and 200 in this line.

Lines 184-186: The discussion regarding C3 and C4 grasses here and/or elsewhere in the manuscript may be enhanced through reference to the following article:

Medek DE, Beggs PJ, Erbas B, Jaggard AK, Campbell BC, Vicendese D, Johnston FH, Godwin I, Huete AR, Green BJ, Burton PK, Bowman DMJS, Newnham RM, Kate-laris CH, Haberle SG, Newbigin E, Davies JM. Regional and seasonal variation in

airborne grass pollen levels between cities of Australia and New Zealand. Aerobiologia 2016;32(2):289-302. DOI: 10.1007/s10453-015-9399-x

Line 278: Change "met" to "meteorological".

Line 343: With respect to the Parry et al. 2007 citation, it should be Confalonieri et al. 2007 because the former are the book editors and the latter are the chapter authors. Change also in the References list (i.e., chapter authors first, book editors later in the reference). Further, here and/or elsewhere in this paragraph (lines 342-352) could benefit from reference to the following:

Ziska LH. Impacts of climate change on allergen seasonality. In: Beggs PJ (Editor). Impacts of Climate Change on Allergens and Allergic Diseases. Cambridge University Press, Cambridge, 2016. pp 92-112.

Ziska L, Knowlton K, Rogers C, Dalan D, Tierney N, Elder MA, Filley W, Shropshire J, Ford LB, Hedberg C, Fleetwood P, Hovanky KT, Kavanaugh T, Fulford G, Vrtis RF, Patz JA, Portnoy J, Coates F, Bielory L, Frenz D. Recent warming by latitude associated with increased length of ragweed pollen season in central North America. Proceedings of the National Academy of Sciences of the United States of America 2011;108(10):4248–4251. DOI: 10.1073/pnas.1014107108

Line 359: This line includes a reference to Table 2. There is no Table 2 in the manuscript. Should the reference be to Table 1?

Lines 505-507, section 5.2.4: This sentence, and this section, seems to neglect any mention that the first of the two observed ragweed peaks in the Mountain subregion (from about day 100 to day 140) is entirely missed by the model.

Note also that the lower row of plots in Figure 12 is mislabelled (except for the first in the row). What are currently r-u should really be q-t. q was missed somehow.

Line 532: See earlier comment regarding Parry et al. 2007. Also, a couple of additional references that would be strong support for this sentence are:

[Figure]

Lake IR, Jones NR, Agnew M, Goodess CM, Giorgi F, Hamaoui-Laguel L, Semenov MA, Solomon F, Storkey J, Vautard R, Epstein MM. Climate change and future pollen allergy in Europe. Environmental Health Perspectives 2017;125(3):385–391. DOI: 10.1289/EHP173

Ziello C, Sparks TH, Estrella N, Belmonte J, Bergmann KC, Bucher E, et al. Changes to airborne pollen counts across Europe. PLoS One 2012;7(4):e34076. DOI: 10.1371/journal.pone.0034076

Line 570: Change "estimating" to "estimation".

Line 584: Change "Association for" to "Academy of".

References: These should be carefully checked to ensure the details and format are correct. Details should be carefully checked against the PDF of each article.

Line 731, Figure 1 caption: Instead of using the word "black" to describe the shading of the Pacific Northwest subregion, perhaps the term "dark grey" would be better.

Figure 2: The RAG and GRA lines are too similar. They are fine when enlarged on screen but when printed they are difficult to tell apart. Perhaps one could be red and the other black (meaning the four lines would be black, red, green, and blue).

Line 733, Figure 2 caption: As stated earlier in the comments with respect to the manuscript as a whole, remove the word "climatological" and replace it with "average", such as: Average daily observed time series of pollen count data . . .

Line 740, Figure 3 caption: BELD is defined as "Biogenic Emissions Landuse Database" in Section 3.1 paragraph 2, not "Biogenic Emissions Land cover Database" as it is here in the figure caption. Which is correct?

Line 748, Figure 4 caption: Indicate that ragweed is "(g)".

Line 757, Figure 6 caption: Change the start of the caption to: Monthly average pollen emissions potential . . .

Lines 777-782, Figure 12 caption: Change the three occurrences of climatological/climatology. The caption can start: Average daily (2003-2010) time series of pollen counts . . .

Line 778: Change RAG from "p-u" to "p-t".

Line 782: Add to the end of the very last sentence "by region and PFT", i.e.: Note: scale of y-axes varies by region and PFT.

Table 1: The numbers in the production factor (P) column should include the same number of numbers after the decimal point (I suggest 1, e.g., 89.1, 210.0, etc.), and the numbers should be aligned right in the column, not aligned left.

―――――――――――――――――――

---

## Referee Comment (RC2) · Anonymous Referee #2 · 8 Aug 2017

The authors have submitted a well-written manuscript describing a prognostic pollen emission model for climate modes. The approach relies on empirical phenological models that have been used extensively in regional scale applications. The novelty of this work arises from using (and comparing) both a taxa and a PFT-specific land cover database to drive pollen emissions in a climate model (RegCM4). The results suggest that the taxa-based model captures in a better way tree-based pollen counts at the expense of losing some of the "climate-flexibility" that the PFT-based model provides.

In addition to the comments and corrections noted by Referee #1, the following specific and general points would enhance the quality of the manuscript:

Line 28: Wind-borne pollen diameters can range more than 70 $\mu$m.

[Figure]

Line 221: BELD is not based only on land surveys. The authors should revise and include version information.

Section 4.3 needs enhancements to better explain how the production factor was obtained for each modeled taxon due to non-uniform methodology. Furthermore, Table 2 does not exist.

Section 5. The regional climate model setup needs to be described in more detail (i.e. number of cells, resolution, vertical structure, etc.).

A taxa-based database comparison providing spatial coverage values for each region would be a useful addition.

References need to be carefully checked - i.e. Zhang, R. et al. (2014)

Production factors and the units listed in Table 1 must be properly referenced.

—————————————————

---

## Author Comment (AC1) · 31 Aug 2017

Response to Referee #1

Thank you for your compliments and constructive suggestions to improve the manuscript. Referee #1's comments are individually listed below with a corresponding author response. Line numbers in author responses correspond to the revised manuscript.

Comment: General: Throughout the manuscript the word "climatological" or "climatology" is used instead of the word "average" or "mean" with regard to pollen. For example, lines 161- 162 state that: "For deciduous broadleaf forest (DBF) taxa, the Southeast has the highest climatological pollen maximum reaching up to about 700-

**GMDD**

grains m-3 around day 100." This is confusing because it is applied to a non climatic/meteorological variable, and because it is used in a manuscript which also focusses on climate. It would be much better to simply state: "For deciduous broadleaf forest (DBF) taxa, the Southeast has the highest average pollen maximum reaching up to about 700-1200 grains m-3 around day 100." Similarly, lines 159-160 could be modified from: "Figure 2 shows the observed clima- tological PFT daily pollen counts averaged over all stations within the defined subre- gions." to: "Figure 2 shows the observed average daily PFT pollen counts averaged over all stations within the defined subregions.". And so on.

Response: All instances of "climatological" or "climatology" with regards to pollen emission fluxes and counts have been changed to "average", sometimes with added specificity (e.g, "8-year average pollen time series", Line 164).

Comment: Line 17, Abstract: "PFT" is used without being given in full earlier in the Abstract, so please provide both the full and abbreviated form here. Response: PFT defined as "plant functional type" at first use in the in abstract (Line 17).

Comment: Line 40, Introduction: The authors may wish to refer to two recently published works that relate to the introductory material here and/or elsewhere in the Introduction: Sofiev M, Prank M. Impacts of climate change on aeroallergen dispersion, transport, and deposition. In: Beggs PJ (Editor). Impacts of Climate Change on Allergens and Allergic Diseases. Cambridge University Press, Cambridge, 2016. pp 50-73. Beggs PJ, Šikoparija B, Smith M. Aerobiology in the International Journal of Biometeorology, 1957–2017. International Journal of Biometeorology 2017. DOI: 10.1007/s00484-017-1374-5 [see section on "Aerobiological modelling and forecasting"]

Response: Beggs et al. 2017 is now cited on a new line that reads, "The interest and growing wealth of knowledge of allergenic pollen is reviewed by Beggs et al. (2017)." (Lines 40-41). Sofiev & Prank 2016 was found useful for the discussion of climate-scale pollen dispersion models, thus it cited in a new line of the introduction, "Only recently have regional-scale modeling studies of pollen dispersion been conducted for Europe, and they have been used to assess the impacts of climate change on airborne pollen distributions." (Lines 45-47).

Comment: Line 98: This sentence makes reference to "the Finnish emergency modeling system (SILAM)". It would be better to change this to "the Finnish System for Integrated modeLling of Atmospheric coMposition (SILAM)" as given in the Introduction section of Sofiev et al. (2013).

Response: This correction to the acronym "SILAM" has been made on Lines 110-111.

Comment: Line 130, paragraph 1 of section 2.1: Table 1 does not relate to "NAB pollen count data ranging from 2003-2010 at all stations in the continental United States", so delete reference to it and just refer to Figure 1.

Response: The reference to Table 1 was intended to be a reference to Table S1; Table S1 is now referenced at the end of paragraph 1 of section 2.1, and any reference to Table 1 has been deleted.

Comment: Line 139: This line includes a reference to Table 2. There is no Table 2 in the manuscript. Should it be Table S2?

Response: The reference to Table 2 was intended to be toward Table S2. It has been corrected on Line 152.

Comment: Line 140: Change "Cupresseceae" to "Cupressaceae".

Response: This spelling correction was made on Line 150.

Comment: Lines 155-159, paragraph 1 of section 2.2: Currently just two of the four boundaries of each of the five subregions are provided. Please provide upper and lower limits of both latitude and longitude for each subregion.

Response: We have revised the subregion boundaries to include full boundaries as displayed in Figure 1. (Lines 167-173).

Comment: Lines 161-170: The paragraph in these lines seems to contain several values that do not match what is shown in Figure 2. Specifically, deciduous broadleaf forest (DBF) taxa in the Southeast does not have an average pollen maximum reaching up to about 700-1200 grains m-3. Figure 2b shows that it only reaches up to about 500 grains m-3. In the Northeast, DBF does not reach up to an average of 400 grains m-3. It peaks just above 240 grains m-3. And finally, a sharp maximum of 775 grains m-3 does not appear in the Mountain subregion. The sharp maximum is only about 360 grains m-3. The paragraph should be carefully checked. Line 173: As above, please check the numbers 400 and 200 in this line.

Response: We thank the reviewer for drawing this to our attention. Figure 2 data was found to have errors, and as a result does not match the text as noted by the reviewer. We have revised Figure 2 with the correct data and now the figure is consistent with the manuscript text.

Comment: Lines 184-186: The discussion regarding C3 and C4 grasses here and/or elsewhere in the manuscript may be enhanced through reference to the following article: Medek DE, Beggs PJ, Erbas B, Jaggard AK, Campbell BC, Vicendese D, Johnston FH, Godwin I, Huete AR, Green BJ, Burton PK, Bowman DMJS, Newnham RM, Katelaris CH, Haberle SG, Newbigin E, Davies JM. Regional and seasonal variation in airborne grass pollen levels between cities of Australia and New Zealand. Aerobiologia 2016;32(2):289-302. DOI: 10.1007/s10453-015-9399-x

Response: Thank you for the suggestion. Medek et al. 2016 has been cited in the main discussion of C3 and C4 grass phenology in Section 2.2. New lines of text are as follows: "Similarly, Medek et al. (2016) observed two grass pollen peaks in Australia, with a stronger, late-summer peak at lower Southern latitudes where there is higher incidence of C4 grass. However, the authors note that sometimes this may be due to a second flowering of some C3 grass species." (Lines 211-213). This paper was also used in the discussion of phenological trends in Section 4.2 with new text as follow: "Trends for grass in Australasia show that the correlation of the end date of the pollen season with average spring temperature is positive, while the same relationship for the start date is negative, suggesting also that season start dates are earlier and season duration increases with warmer climates (Medek et al. 2016)." (Lines 408-410).

Comment: Line 278: Change "met" to "meteorological".

Response: Correction made from "met" to "meteorological". (Line 339). Comment: Line 343: With respect to the Parry et al. 2007 citation, it should be Confalonieri et al. 2007 because the former are the book editors and the latter are the chapter authors. Change also in the References list (i.e., chapter authors first, book editors later in the reference). Further, here and/or elsewhere in this paragraph (lines 342-352) could benefit from reference to the following: Ziska LH. Impacts of climate change on allergen seasonality. In: Beggs PJ (Editor). Impacts of Climate Change on Allergens and Allergic Diseases. Cambridge University Press, Cambridge, 2016. pp 92-112. Ziska L, Knowlton K, Rogers C, Dalan D, Tierney N, Elder MA, Filley W, Shropshire J, Ford LB, Hedberg C, Fleetwood P, Hovanky KT, Kavanaugh T, Fulford G, Vrtis RF, Patz JA, Portnoy J, Coates F, Bielory L, Frenz D. Recent warming by latitude associated with increased length of ragweed pollen season in central North America. Proceedings of the National Academy of Sciences of the United States of America 2011;108(10):4248–4251. DOI: 10.1073/pnas.1014107108

Response: The reference to Parry et al. 2007 has been changed to Confalonieri et al. 2007 as suggested. The additional references are excellent examples of general and advanced discussions on pollen phenology. Ziska 2016 is cited several times throughout the manuscript : "The spatiotemporal heterogeneity of climate change may affect which regions and seasons will be most influenced by climate change (Ziska 2016)." (Line 415-416), with a second citation on Line 419-420 and a third citation in the introduction: "Climatic changes in large-scale pollen distributions are mostly absent from scientific literature, though multiple studies on phenological changes in the pollen season have been published" (Lines 43-44). Ziska et al. 2011 has been cited in the discussion of ragweed phenology in Section 4.2: "The apparent trend in the season end date for Ambrosia with PYAAT could be due to the increased number of frost-free days, consistent with global warming, and a strong relationship between frost-free days and changes of ragweed season length" (Lines 410-413).

Comment: Line 359: This line includes a reference to Table 2. There is no Table 2 in the manuscript. Should the reference be to Table 1?

Reponse: We apologize for this mistake. The table reference in section 4.3 should be to Table 1. This has been corrected.

Comment: Lines 505-507, section 5.2.4: This sentence, and this section, seems to neglect any mention that the first of the two observed ragweed peaks in the Mountain subregion (from about day 100 to day 140) is entirely missed by the model.

Response: We have added a note about the spring ragweed peak in the Mountain sub-region in the discussion ("There is a yet unidentified observed spring peak of ragweed pollen at about day 125 in the Mountain subregion, possibly due to an identification error." lines 641-642), though no known source can yet be identified for these somewhat unusual ragweed pollen counts.

Comment: Note also that the lower row of plots in Figure 12 is mislabelled (except for the first in the row). What are currently r-u should really be q-t. q was missed somehow.

Response: Thank you for pointing out this error. Figure 12 has been updated with corrected panel labels, now including q and excluding u.

Comment: Line 532: See earlier comment regarding Parry et al. 2007. Also, a couple of additional references that would be strong support for this sentence are: Lake IR, Jones NR, Agnew M, Goodess CM, Giorgi F, Hamaoui-Laguel L, Semenov MA, Solomon F, Storkey J, Vautard R, Epstein MM. Climate change and future pollen allergy in Europe. Environmental Health Perspectives 2017;125(3):385–391. DOI:

10.1289/EHP173 Ziello C, Sparks TH, Estrella N, Belmonte J, Bergmann KC, Bucher E, et al. Changes to airborne pollen counts across Europe. PLoS One 2012;7(4):e34076. DOI: 10.1371/journal.pone.0034076

Response: Additional references Ziello et al. 2012 and Lake et al. 2017 have been added to the citation in Line 664 as they support this discussion point. In addition, Lake et al. 2017 was cited in Lines 45-47 in the discussion of recent pollen dispersion modeling efforts.

Comments: Line 570: Change "estimating" to "estimation".

Response: Change made on Line 703.

Comment: Line 584: Change "Association for" to "Academy of".

Response: Change made on Line 718.

Comment: References: These should be carefully checked to ensure the details and format are correct. Details should be carefully checked against the PDF of each article.

Response: All references have been checked against their articles, and the details have been corrected for any references that were missing components or incorrectly formatted.

Comment: Line 731, Figure 1 caption: Instead of using the word "black" to describe the shading of the Pacific Northwest subregion, perhaps the term "dark grey" would be better.

Response: "black" updated to "dark grey" in Figure 1 caption as suggested.

Comment: Figure 2: The RAG and GRA lines are too similar. They are fine when enlarged on screen but when printed they are difficult to tell apart. Perhaps one could be red and the other black (meaning the four lines would be black, red, green, and blue).

[Figure]

Response: The line colors in Figure 2 have been updated such that the previously red ragweed line is now magenta. This appears to provide good contrast for all lines in this plot.

Comment: Line 733, Figure 2 caption: As stated earlier in the comments with respect to the manuscript as a whole, remove the word "climatological" and replace it with "average", such as: Average daily observed time series of pollen count data . . .

Response: This language has been updated from "climatological" to "average".

Comment: Line 740, Figure 3 caption: BELD is defined as "Biogenic Emissions Landuse Database" in Section 3.1 paragraph 2, not "Biogenic Emissions Land cover Database" as it is here in the figure caption. Which is correct?

Response: We apologize for this confusion. The correct name is "Biogenic Emissions Landuse Database". The Figure 3 caption has been updated to reflect this.

Comment: Line 748, Figure 4 caption: Indicate that ragweed is "(g)".

Response: Figure 4 caption has been updated accordingly.

Comment: Line 757, Figure 6 caption: Change the start of the caption to: Monthly average pollen emissions potential . . .

Response: This correction has been made to the Figure 6 caption.

Comment: Lines 777-782, Figure 12 caption: Change the three occurrences of climatologi- cal/climatology. The caption can start: Average daily (2003-2010) time series of pollen counts . . .

Response: The Figure 12 caption has been updated accordingly.

Comment: Line 778: Change RAG from "p-u" to "p-t".

Response: The Figure 12 caption has been updated to reflect the changes to the panel labels in the figure.

Comment: Line 782: Add to the end of the very last sentence "by region and PFT", i.e.: Note: scale of y-axes varies by region and PFT.

Response: This addition has been made to the Figure 12 caption.

Comment: Table 1: The numbers in the production factor (P) column should include the same number of numbers after the decimal point (I suggest 1, e.g., 89.1, 210.0, etc.), and the numbers should be aligned right in the column, not aligned left.

Response: All values right aligned. Decimal places are left unchanged for production factors as those are formatted to reflect the correct number of significant digits in that data. Linear regression values are also updated so that all numbers are rounded to two decimal places.

Best,

Matthew Wozniak

Please also note the supplement to this comment:
https://www.geosci-model-dev-discuss.net/gmd-2017-105/gmd-2017-105-AC1-supplement.pdf

**Supplement:**

**Response to Referee #1**

Thank you for your compliments and constructive suggestions to improve the manuscript. Referee #1's comments are individually listed below with a corresponding author response. Line numbers in author responses correspond to the revised manuscript.

**Comment:** General: Throughout the manuscript the word "climatological" or "climatology" is used instead of the word "average" or "mean" with regard to pollen. For example, lines 161- 162 state that: "For deciduous broadleaf forest (DBF) taxa, the Southeast has the highest climatological pollen maximum reaching up to about 700-1200 grains m-3 around day 100." This is confusing because it is applied to a non climatic/meteorological variable, and because it is used in a manuscript which also focusses on climate. It would be much better to simply state: "For deciduous broadleaf forest (DBF) taxa, the Southeast has the highest average pollen maximum reaching up to about 700-1200 grains m-3 around day 100."
Similarly, lines 159-160 could be modified from: "Figure 2 shows the observed clima- tological PFT daily pollen counts averaged over all stations within the defined subre- gions." to: "Figure 2 shows the observed average daily PFT pollen counts averaged over all stations within the defined subregions.".
And so on.

**Response:** All instances of "climatological" or "climatology" with regards to pollen emission fluxes and counts have been changed to "average", sometimes with added specificity (e.g, "8-year average pollen time series", Line 164).

**Comment**: Line 17, Abstract: "PFT" is used without being given in full earlier in the Abstract, so please provide both the full and abbreviated form here.

**Response:** PFT defined as "plant functional type" at first use in the in abstract (Line 17).

**Comment:** Line 40, Introduction: The authors may wish to refer to two recently published works that relate to the introductory material here and/or elsewhere in the Introduction:
Sofiev M, Prank M. Impacts of climate change on aeroallergen dispersion, transport, and deposition. In: Beggs PJ (Editor). Impacts of Climate Change on Allergens and Allergic Diseases. Cambridge University Press, Cambridge, 2016. pp 50-73.
Beggs PJ, Šikoparija B, Smith M. Aerobiology in the International Journal of Biometeorology, 1957–2017. International Journal of Biometeorology 2017. DOI: 10.1007/s00484-017-1374-5 [see section on "Aerobiological modelling and forecasting"]

**Response**: Beggs et al. 2017 is now cited on a new line that reads, "The interest and growing wealth of knowledge of allergenic pollen is reviewed by *Beggs et al.* (2017)." (Lines 40-41).
Sofiev & Prank 2016 was found useful for the discussion of climate-scale pollen dispersion models, thus it cited in a new line of the introduction, "Only recently have regional-scale modeling studies of pollen dispersion been conducted for Europe, and they have been used to assess the impacts of climate change on airborne pollen distributions." (Lines 45-47).

**Comment:** Line 98: This sentence makes reference to "the Finnish emergency modeling system (SILAM)". It would be better to change this to "the Finnish System for Integrated modeLling of Atmospheric coMposition (SILAM)" as given in the Introduction section of Sofiev et al. (2013).

**Response:** This correction to the acronym "SILAM" has been made on Lines 110-111.

**Comment:** Line 130, paragraph 1 of section 2.1: Table 1 does not relate to "NAB pollen count data ranging from 2003-2010 at all stations in the continental United States", so delete reference to it and just refer to Figure 1.

**Response:** The reference to Table 1 was intended to be a reference to Table S1; Table S1 is now referenced at the end of paragraph 1 of section 2.1, and any reference to Table 1 has been deleted.

**Comment:** Line 139: This line includes a reference to Table 2. There is no Table 2 in the manuscript. Should it be Table S2?

**Response:** The reference to Table 2 was intended to be toward Table S2. It has been corrected on Line 152.

**Comment:** Line 140: Change "Cupresseceae" to "Cupressaceae".

**Response:** This spelling correction was made on Line 150.

**Comment:** Lines 155-159, paragraph 1 of section 2.2: Currently just two of the four boundaries of each of the five subregions are provided. Please provide upper and lower limits of both latitude and longitude for each subregion.

**Response:** We have revised the subregion boundaries to include full boundaries as displayed in Figure 1. (Lines 167-173).

**Comment**: Lines 161-170: The paragraph in these lines seems to contain several values that do not match what is shown in Figure 2. Specifically, deciduous broadleaf forest (DBF) taxa in the Southeast does not have an average pollen maximum reaching up to about 700-1200 grains m-3. Figure 2b shows that it only reaches up to about 500 grains m-3. In the Northeast, DBF does not reach up to an average of 400 grains m-3. It peaks just above 240 grains m-3. And finally, a sharp maximum of 775 grains m-3 does not appear in the Mountain subregion. The sharp maximum is only about 360 grains m-3. The paragraph should be carefully checked.
Line 173: As above, please check the numbers 400 and 200 in this line.

**Response:** We thank the reviewer for drawing this to our attention. Figure 2 data was found to have errors, and as a result does not match the text as noted by the reviewer. We have revised Figure 2 with the correct data and now the figure is consistent with the manuscript text.

**Comment:** Lines 184-186: The discussion regarding C3 and C4 grasses here and/or elsewhere in the manuscript may be enhanced through reference to the following article:
Medek DE, Beggs PJ, Erbas B, Jaggard AK, Campbell BC, Vicendese D, Johnston FH, Godwin I, Huete AR, Green BJ, Burton PK, Bowman DMJS, Newnham RM, Kate- laris CH, Haberle SG, Newbigin E, Davies JM. Regional and seasonal variation in airborne grass pollen levels between cities of Australia and New Zealand. Aerobiologia 2016;32(2):289-302. DOI: 10.1007/s10453-015-9399-x

**Response:** Thank you for the suggestion. Medek et al. 2016 has been cited in the main discussion of $C_3$ and $C_4$ grass phenology in Section 2.2. New lines of text are as follows: "Similarly, *Medek et al.* (2016) observed two grass pollen peaks in Australia, with a stronger, late-summer peak at lower Southern latitudes where there is higher incidence of $C_4$ grass. However, the authors note that sometimes this may be due to a second flowering of some $C_3$ grass species." (Lines 211-213). This paper was also used in the discussion of phenological trends in Section 4.2 with new text as follow: "Trends for grass in Australasia show that the correlation of the end date of the pollen season with average spring temperature is positive, while the same relationship for the start date is negative, suggesting also that season start dates are earlier and season duration increases with warmer climates (Medek et al. 2016)." (Lines 408-410).

**Comment:** Line 278: Change "met" to "meteorological".

**Response:** Correction made from "met" to "meteorological". (Line 339).

**Comment:** Line 343: With respect to the Parry et al. 2007 citation, it should be Confalonieri et al. 2007 because the former are the book editors and the latter are the chapter authors. Change also in the References list (i.e., chapter authors first, book editors later in the reference). Further, here and/or elsewhere in this paragraph (lines 342-352) could benefit from reference to the following:

Ziska LH. Impacts of climate change on allergen seasonality. In: Beggs PJ (Editor). Impacts of Climate Change on Allergens and Allergic Diseases. Cambridge University Press, Cambridge, 2016. pp 92-112.

Ziska L, Knowlton K, Rogers C, Dalan D, Tierney N, Elder MA, Filley W, Shropshire J, Ford LB, Hedberg C, Fleetwood P, Hovanky KT, Kavanaugh T, Fulford G, Vrtis RF, Patz JA, Portnoy J, Coates F, Bielory L, Frenz D. Recent warming by latitude associated with increased length of ragweed pollen season in central North America. Proceedings of the National Academy of Sciences of the United States of America 2011;108(10):4248– 4251. DOI: 10.1073/pnas.1014107108

**Response:** The reference to Parry et al. 2007 has been changed to Confalonieri et al. 2007 as suggested. The additional references are excellent examples of general and advanced discussions on pollen phenology. Ziska 2016 is cited several times throughout the manuscript : **"**The spatiotemporal heterogeneity of climate change may affect which regions and seasons will be most influenced by climate change (Ziska 2016)." (Line 415-416), with a second citation on Line 419-420 and a third citation in the introduction: "Climatic changes in large-scale pollen distributions are mostly absent from scientific literature, though multiple studies on phenological changes in the pollen season have been published" (Lines 43-44).
Ziska et al. 2011 has been cited in the discussion of ragweed phenology in Section 4.2: "The apparent trend in the season end date for *Ambrosia* with PYAAT could be due to the increased number of frost-free days, consistent with global warming, and a strong relationship between frost-free days and changes of ragweed season length" (Lines 410-413).

**Comment:** Line 359: This line includes a reference to Table 2. There is no Table 2 in the manuscript. Should the reference be to Table 1?

**Reponse**: We apologize for this mistake. The table reference in section 4.3 should be to Table 1. This has been corrected.

**Comment:** Lines 505-507, section 5.2.4: This sentence, and this section, seems to neglect any mention that the first of the two observed ragweed peaks in the Mountain subregion (from about day 100 to day 140) is entirely missed by the model.

**Response:** We have added a note about the spring ragweed peak in the Mountain subregion in the discussion ("There is a yet unidentified observed spring peak of ragweed pollen at about day 125 in the Mountain subregion, possibly due to an identification error." lines 641-642), though no known source can yet be identified for these somewhat unusual ragweed pollen counts.

**Comment:** Note also that the lower row of plots in Figure 12 is mislabelled (except for the first in the row). What are currently r-u should really be q-t. q was missed somehow.

**Response:** Thank you for pointing out this error. Figure 12 has been updated with corrected panel labels, now including q and excluding u.

**Comment:** Line 532: See earlier comment regarding Parry et al. 2007. Also, a couple of additional references that would be strong support for this sentence are:
Lake IR, Jones NR, Agnew M, Goodess CM, Giorgi F, Hamaoui-Laguel L, Semenov MA, Solomon F, Storkey J, Vautard R, Epstein MM. Climate change and future pollen allergy in Europe. Environmental Health Perspectives 2017;125(3):385–391. DOI: 10.1289/EHP173
Ziello C, Sparks TH, Estrella N, Belmonte J, Bergmann KC, Bucher E, et al. Changes to airborne pollen counts across Europe. PLoS One 2012;7(4):e34076. DOI: 10.1371/journal.pone.0034076

**Response:** Additional references Ziello et al. 2012 and Lake et al. 2017 have been added to the citation in Line 664 as they support this discussion point. In addition, Lake et al. 2017 was cited in Lines 45-47 in the discussion of recent pollen dispersion modeling efforts.

**Comments:** Line 570: Change "estimating" to "estimation".

**Response:** Change made on Line 703.

**Comment:** Line 584: Change "Association for" to "Academy of".

**Response:** Change made on Line 718.

**Comment:** References: These should be carefully checked to ensure the details and format are correct. Details should be carefully checked against the PDF of each article.

**Response:** All references have been checked against their articles, and the details have been corrected for any references that were missing components or incorrectly formatted.

**Comment:** Line 731, Figure 1 caption: Instead of using the word "black" to describe the shading of the Pacific Northwest subregion, perhaps the term "dark grey" would be better.

 **Response:** "black" updated to "dark grey" in Figure 1 caption as suggested.

**Comment:** Figure 2: The RAG and GRA lines are too similar. They are fine when enlarged on screen but when printed they are difficult to tell apart. Perhaps one could be red and the other black (meaning the four lines would be black, red, green, and blue).

**Response:** The line colors in Figure 2 have been updated such that the previously red ragweed line is now magenta. This appears to provide good contrast for all lines in this plot.

**Comment:** Line 733, Figure 2 caption: As stated earlier in the comments with respect to the manuscript as a whole, remove the word "climatological" and replace it with "average", such as: Average daily observed time series of pollen count data . . .

**Response:** This language has been updated from "climatological" to "average".

**Comment:** Line 740, Figure 3 caption: BELD is defined as "Biogenic Emissions Landuse Database" in Section 3.1 paragraph 2, not "Biogenic Emissions Land cover Database" as it is here in the figure caption. Which is correct?

**Response:** We apologize for this confusion. The correct name is "Biogenic Emissions Landuse Database". The Figure 3 caption has been updated to reflect this.

**Comment:** Line 748, Figure 4 caption: Indicate that ragweed is "(g)".

**Response:** Figure 4 caption has been updated accordingly.

**Comment:** Line 757, Figure 6 caption: Change the start of the caption to: Monthly average pollen emissions potential . . .

**Response:** This correction has been made to the Figure 6 caption.

**Comment:** Lines 777-782, Figure 12 caption: Change the three occurrences of climatological/climatology. The caption can start: Average daily (2003-2010) time series of pollen counts . . .

**Response:** The Figure 12 caption has been updated accordingly.

**Comment:** Line 778: Change RAG from "p-u" to "p-t".

**Response:** The Figure 12 caption has been updated to reflect the changes to the panel labels in the figure.

**Comment:** Line 782: Add to the end of the very last sentence "by region and PFT", i.e.: Note: scale of y-axes varies by region and PFT.

**Response:** This addition has been made to the Figure 12 caption.

**Comment:** Table 1: The numbers in the production factor (P) column should include the same number of numbers after the decimal point (I suggest 1, e.g., 89.1, 210.0, etc.), and the numbers should be aligned right in the column, not aligned left.

**Response:** All values right aligned. Decimal places are left unchanged for production factors as those are formatted to reflect the correct number of significant digits in that data. Linear regression values are also updated so that all numbers are rounded to two decimal places.

Best,

Matthew Wozniak

**A prognostic pollen emissions model for climate models (PECM1.0)**

Matthew C. Wozniak[1], Allison L. Steiner[1]

[1]Climate and Space Sciences and Engineering, University of Michigan, Ann Arbor, MI 48109, USA

*Correspondence to*: Matthew C. Wozniak (mcwoz@umich.edu)

**Abstract.** We develop a prognostic model of Pollen Emissions for Climate Models (PECM) for use within regional and global climate models to simulate pollen counts over the seasonal cycle based on geography, vegetation type and meteorological parameters. Using modern surface pollen count data, empirical relationships between prior-year annual average temperature and pollen season start dates and end dates are developed for deciduous broadleaf trees (*Acer, Alnus, Betula, Fraxinus, Morus, Platanus, Populus, Quercus, Ulmus*), evergreen needleleaf trees (Cupressaceae, Pinaceae), grasses (Poaceae; $C_3$, $C_4$), and ragweed (*Ambrosia*). This regression model explains as much as 57% of the variance in pollen phenological dates, and it is used to create a "climate-flexible" phenology that can be used to study the response of wind-driven pollen emissions to climate change. The emissions model is evaluated in a regional climate model (RegCM4) over the continental United States by prescribing an emission potential from PECM and transporting pollen as aerosol tracers. We evaluate two different pollen emissions scenarios in the model, using: (1) a taxa-specific land cover database, phenology and emission potential, and (2) a plant functional type (PFT) land cover, phenology and emission potential. The simulated surface pollen concentrations for both simulations are evaluated against observed surface pollen counts in five climatic subregions. Given prescribed pollen emissions, the RegCM4 simulates observed concentrations within an order of magnitude, although the performance of the simulations in any subregion is strongly related to the land cover representation and the number of observation sites used to create the empirical phenological relationship. The taxa-based model provides a better representation of the phenology of tree-based pollen counts than the PFT-based model, however we note that the PFT-based version provides a useful and "climate-flexible" emissions model for the general representation of the pollen phenology over the United States.

**1 Introduction**

Pollen grains are released from plants to transmit the male genetic material for reproduction. When lofted into the atmosphere, they represent a natural source of coarse atmospheric aerosols, ranging typically from 15 to 60 μm in diameter, while sometimes exceeding 100 μm (Cecchi 2014; Sofiev et al. 2014). In the mid-latitudes, much of the vegetation relies dominantly on anemophilous, or wind-driven, pollination (Lewis et al. 1983), representing a closely coupled relationship of pollen emissions to weather and climate. Anemophilous pollinators include woody plants such as trees and shrubs, as well as other non-woody vascular plants such as grasses and herbs. Pollen emissions are directly affected by meteorological (e.g., temperature, wind, relative humidity) and climatological (e.g., temperature, soil moisture) factors (Weber 2003). Aerobiology studies indicate that after release, pollen can be transported on the order of ten to a thousand kilometers (Sofiev et al. 2006; Schueler and Schlünzen 2006; Kuparinen et al. 2007) but there are still large uncertainties regarding emissions and transport of pollen.

Prognostic pollen emissions are useful for the scientific community and public, specifically for forecasting allergenic conditions or predicting the flow of genetic material. The interest and growing wealth of knowledge of allergenic pollen has been recently reviewed by *Beggs et al.* (2017). To date, most pollen emissions models focus on relatively short, seasonal time scales and smaller locales for a limited selection of taxa (Sofiev et al. 2013; Liu et al. 2016; R. Zhang et al. 2014). Climatic changes in large-scale pollen distributions are mostly absent from scientific literature, though multiple studies on phenological changes in the pollen season have been published (Ziska 2016; Yue et al. 2015; Y. Zhang et al. 2015a). Only recently have regional-scale modeling studies of pollen dispersion been conducted for Europe, and they have been used to assess the impacts of climate change on airborne pollen distributions (Sofiev and Prank 2016; Lake 
[revised manuscript text omitted]

and 70°-100°W; 34 stations), the Southeast (temperate, subtropical; 25-38°N and 70°-100°W; 29 stations), Mountain (varied climate; 25°-48°N and 100°-116°W; 9 stations), California (Mediterranean, varied climate; 25°-40°N and

116°-125°W; 13 stations) and the Pacific Northwest (temperate rainforest; west of 116°W and north of 40°N; 4

stations). Figure 2 shows the observed average PFT daily pollen counts averaged over all stations within the defined subregions.

For deciduous broadleaf forest (DBF) taxa, the Southeast has the highest average pollen maximum reaching up to about 700-1200 grains m$^{-3}$ around day 100.  In the Northeast, DBF is the dominant PFT, reaching up to an average of 400 grains m$^{-3}$ and peaking slightly later (around day 120) than the Southeast. California sites show an average peak around 150 grains m$^{-3}$ occurring slightly earlier around day 80. A sharp maximum of 775 grains m$^{-3}$ appears in the Mountain subregion at about day 80, with a secondary emission reaching around 150 grains m$^{-3}$ on day 125.  In the Northwest, DBF pollen has the earliest maximum (day 70) at about the same magnitude as California (~200

grains m$^{-3}$). In some locations, there is a secondary DBF peak in the late summer and early fall due to the late flowering of *Ulmus crassifolia* and *Ulmus parvifolia*, located predominantly in the Southeast and California (Lewis,

Matthew Wozniak 8/2/2017 6:07 PM

Matthew Wozniak 8/2/2017 5:14 PM

Matthew Wozniak 8/9/2017 11:06 AM

Matthew Wozniak 8/9/2017 10:55 AM

Matthew Wozniak 8/9/2017 10:54 AM

Matthew Wozniak 8/9/2017 10:54 AM

Matthew Wozniak 8/9/2017 10:55 AM

Matthew Wozniak 8/9/2017 10:56 AM

Matthew Wozniak 8/9/2017 10:56 AM

Matthew Wozniak 8/9/2017 10:57 AM

Matthew Wozniak 8/9/2017 10:57 AM

Matthew Wozniak 8/2/2017 5:17 PM

Matthew Wozniak 8/2/2017 5:17 PM

Matthew Wozniak 8/2/2017 5:18 PM

et al.1983). In the Southeast this occurs between days 225 and 300, while in California this occurs twice around day

245 and day 265.

The two ENF families exhibit pollen release at two distinct but overlapping times, with Cupressaceae peaking before

Pinaceae. Cupressaceae in the Southeast emits pollen earlier than in other subregions, with a maxima at just over

400 grains $m^{-3}$ around day 10 and counts above 200 grains $m^{-3}$ in December of the prior year. Cupressaceae dominates the total emissions for the Southeast, with a smaller maximum from Pinaceae of about 180 grains $m^{-3}$

near day 110. In the Northeast, the bimodality of ENF is evident with the Cupressacaeae family reaching a maximum of 100 grains $m^{-3}$ near day 85 with a secondary Pinaceae maximum approximately 65 days later at about half the magnitude (~50 grains $m^{-3}$). In the Mountain and Pacific Northwest subregions, the maximum occurs around day 50-80 and can reach up to 350 grains $m^{-3}$ in the Mountain subregion, but in both subregions is generally much lower than the eastern United States (approximately 50 grains $m^{-3}$). In the California subregion, ENF emissions are comparatively low (< 50 grains $m^{-3}$) which is likely due to the bias in sampling locations.

The grasses (Poaceae) have comparatively low average pollen counts (<25 grains $m^{-3}$) throughout the season in all subregions except the Northwest, where the maximum reaches 75 grains $m^{-3}$. However, the average maximum

Poaceae pollen count at individual stations is close to 100 grains $m^{-3}$, with the individual annual maxima reaching several hundreds of pollen grains $m^{-3}$. In the AAAAI data, there are two distinct maxima in the Northeast Poaceae count, and we attribute the first seasonal maximum to C$_3$ grasses (peak around day 155) and the second grass maximum mainly to C$_4$ grasses (peak around day 250). Observations by *Craine et al.* (2011) of Poaceae in an

American prairie have indicated that C$_3$ and C$_4$ grass flowering occurs at distinctly different times, with C$_3$ in the late spring and C$_4$ in mid- to late summer. Similarly, *Medek et al.* (2016) observed two grass pollen peaks in

Australia, with a stronger, late-summer peak at lower Southern latitudes where there is higher incidence of C$_4$ grass.

However, the authors note that sometimes this may be due to a second flowering of some C$_3$ grass species. Although the C$_3$-C$_4$ separation cannot be confirmed in the AAAAI pollen count data because they are not distinguished during pollen identification, this distinction is included in the model as discussed in Sections 3.1 and 4.2 below. In the

[revised manuscript text omitted]
). Prior studies have shown that the meteorology of the year previous to the pollen season influences pollen production, especially temperature, suggesting that PYAAT may be a good predictor variable (Menzel and Jochner 2016). While emissions in this study are calculated using offline meteorological data, this also could be coupled to a dynamic land surface model to predict reasonably accurate pollen phenological dates.

To exemplify this method, Figure 5 shows the phenological dates and regression lines for the *Betula* (birch) genus, with all 13 modeled taxon shown in Figures S1 and S2. The sDOY and eDOY of the pollen season show a moderate and considerable trend with temperature for most taxa and PFTs (Table 1; Figures S1 & S2). The linear regression models for sDOY explain 41% of the variance on average for DBF taxa, 47% on average for ENF taxa, 48% for $C_3$ Poaceae, and 8% for *Ambrosia* while having a negligible $R^2$ for $C_4$ Poaceae. For eDOY, the linear regression models explain 21% of the variance on average for DBF taxa, 29% for ENF, 4% for $C_3$ Poaceae, 32% for $C_4$ Poaceae, and 37% for *Ambrosia*. All trends except $C_4$ Poaceae, late elm, and *Ambrosia* are negative, indicating that warmer previous-year temperatures result in earlier start and end dates. For most tree taxa, the trend of both sDOY and eDOY are negatively correlated with PYAAT, with a steeper negative slope for sDOY. The correlation for the duration of the pollen season (eDOY – sDOY) is then positive for all taxa except Cupressaceae. This suggests that warmer climates have earlier pollen season start and end dates but longer season lengths.

Trends for grass in Australasia show that the correlation of the end date of the pollen season with average spring temperature is positive, while the same relationship for the start date is negative, suggesting also that season start dates are earlier and season duration increases with warmer climates (Medek et al. 2016). The apparent trend in the season end date for *Ambrosia* with PYAAT could be due to the increased number of frost-free days, consistent with global warming, and a strong relationship between frost-free days and changes of ragweed season length (Easterling 2002; Ziska et al. 2011).

This agrees with earlier findings that suggest the pollen season will, on average, start earlier with a warmer global climate and have a longer duration (Confalonieri et al. 2007). The spatiotemporal heterogeneity of climate change may affect which regions and seasons will be most influenced by climate change (Ziska 2016). In fact, there is imperfect agreement that earlier start dates and longer seasons will occur unanimously throughout the United States region, at least for trees (Yue et al. 2015). It is understood that photoperiod and the dormancy-breaking process controlled by chilling temperatures play a significant role in the phenology of trees (Myking and Heide 1995; Ziska 2016), and it is generally accepted that a plethora of other factors, such as plant age, mortality, and nutrient availability also affect observed phenological dates (Jochner et al. 2013). However, even without these factors, the current phenological model is applicable to large regions and provides a clear response of plants to inter-annual climate variability as well as long-term climate changes. For this first assessment of PECM, we assume that the pollen production factor ($p_{annual}$) does not change with time and that the phenological model described above captures the main features of pollen emissions.

**4.3 Annual pollen production ($p_{annual}$)**

Annual production factors (grains $m^{-2}$ $year^{-1}$, where $m^{-2}$ refers to vegetated area, or grains $stem^{-1}$ $year^{-1}$ for ragweed) for each modeled taxon are provided in Table 1. The annual pollen production factor ($p_{annual}$) defines the amount of pollen produced per vegetation biomass per year based on literature values. *Tormo* Molina et al. (1996) report the annual pollen productivity in grains tree$^{-1}$ year$^{-1}$ measured from three representative trees from several taxa. *Morus* has no known reference for production factor and was assumed to be $10x10^7$ grains m$^{-2}$ year$^{-1}$, conservatively at the low end of the range for other deciduous broadleaf taxa. Other tree taxa and grasses are reported in grains m$^{-2}$ year$^{-1}$, while ragweed is reported in grains stem$^{-1}$ year$^{-1}$ (Helbig et al. 2004; Jato, Rodríguez-Rajo, and Aira 2007; Hidalgo, Galán, and Domínguez 1999; Prieto-Baena et al. 2003; Fumanal, Chauvel, and Bretagnolle 2007). To convert the production factors from *Tormo Molina et al.* (1996) (grains tree$^{-1}$ year$^{-1}$), the production factors for each representative tree are multiplied by the tree crown area, calculated as the circular area of the tree crown diameter given in Table II of *Tormo Molina et al.* (1996). The resulting individual production factors (grains m$^{-2}$ year$^{-1}$) are then averaged for each taxa.

[revised manuscript text omitted]

Model version 4 (Giorgi et al. 2012), which is a limited-area climate model that includes a coupled aerosol tracer module (Solmon et al. 2006) that readily accommodates pollen tracers (Liu et al. 2016). The pollen tracer transport scheme is extended from one to four bins in this study to simulate the four PFTs (DBF, ENF, GRA, and RAG), with tracer bin particle effective diameters of 28 μm, 40 μm, 35 μm and 20μm, respectively. Additionally, the temporal emissions input is updated to accommodate daily pollen emissions (grains m$^{-2}$ day$^{-1}$).

RegCM4 is based on the hydrostatic version of the Penn State/NCAR mesoscale model MM5 (Grell et al. 1994) and configured for long-term climate simulations. In our RegCM4 configuration, we use the Community Land Model version 4.5 (CLM4.5; (Oleson et al. 2010)), the Emanuel cumulus precipitation scheme over land and ocean (Emanuel 1991), and the SUBEX resolvable scale precipitation (Pal et al. 2000). The horizontal resolution is 25-km with 144x243 grid cells on a Lambert Conformal Projection centered on 39ºN, 100ºW with parallels at 30ºN and

60ºN (Figure 1). The vertical resolution includes 18 vertical sigma levels. Boundary conditions are driven by ERA-

Interim Reanalysis while sea surface temperatures are prescribed from NOAA Optimum Interpolation SSTs (Dee et al. 2011; Smith et al. 2008). Two 8-year simulations of pollen emissions and transport in RegCM4 were conducted from 2003-2010 with the BELD and PFT version of the offline emissions model. Six months of spin-up (July-

[revised manuscript text omitted]

Southeast, (c) Mountain, (d) California, and (e) Pacific Northwest.

[Figure]

[Figure]

**Figure 3.** Land cover fraction (% coverage) for 11 tree taxa from the Biogenic Emissions Landuse Database (BELD3) regridded to a 25km resolution grid, including: a) *Acer* (maple), b) *Alnus* (alder), c) *Betula* (birch), d) Cupressaceae (cedar/juniper), e) *Fraxinus* (ash), f) *Morus* (mulberry), g) Pinaceae (pine), h) *Platanus* (sycamore), i) *Populus* (poplar/aspen), j) *Quercus* (oak), k) *Ulmus* (elm).

Matthew Wozniak 8/10/2017 2:57 PM

Matthew Wozniak 8/2/2017 7:30 PM

[Figure]

**Figure 4.** BELD3 (a, c) and CLM4 (b, d, e, f) land cover for the four PFT categories that produce pollen emissions, including (1) deciduous broadleaf forest for (a) BELD3 and (b) CLM4, (2) evergreen needleaf forest for (c) BELD3

and (d) CLM4, (3) grasses, including (e) C3 grasses and (f) C4 grasses, and (g) ragweed, represented by crop and urban CLM4 categories.

Matthew Wozniak 8/10/2017 3:06 PM

Matthew Wozniak 8/10/2017 3:06 PM

Matthew Wozniak 8/10/2017 3:06 PM

Matthew Wozniak 8/2/2017 7:31 PM

Matthew Wozniak 8/10/2017 3:06 PM

[Figure]

**Figure 5.** Phenological regressions for *Betula* (birch) pollen for (a) Start Day of Year (sDOY) and (b) End Day of

Year (eDOY) versus previous year annual average temperature (PYAAT; ºC). Each point signifies one station per year for pollen count data from 2003-2010 (total denoted as N).

[Figure]

**Figure 6.** Monthly average emissions potential (E; Equation 1) for BELD model DBF (2003-2010), in grains m$^{-2}$

day$^{-1}$. a) January, b) February, c) March, d) April, e) May, f) June, g) July, h) August, i) September, j) October, k)

November, l) December.

Matthew Wozniak 8/2/2017 7:32 PM

[Figure]

**Figure 7.** Same as Figure 6, but for PFT model DBF.

[Figure]

**Figure 8.** Same as Figure 6, but for BELD model ENF.

[Figure]

**Figure 9.** Same as Figure 6, but for PFT model ENF.

[Figure]

**Figure 10.** Same as Figure 6, but for $C_3 + C_4$ grass.

[Figure]

**Figure 11.** Same as Figure 6, but for ragweed.

[Figure]

**Figure 12.** Average (2003-2010) time series of daily pollen counts comparing model and observations for four PFTs (a-e, deciduous broadleaf, DBF; f-j, evergreen needleleaf, ENF; k-o, grasses, GRA; p-t, ragweed, RAG) across 5 U.S. subregions (columns from left to right: Northeast, NE; Southeast, SE; Mountain, MT; California; Pacific Northwest, PNW). Shading for the observations and model represents the mean absolute deviation from the average for each day of the time series. Note: scale of y-axes varies by region and PFT.

[Figure]

**Figure 13.** Box-and-whisker plots showing the statistical spread of the pollen count magnitudes from the regional averages presented in Figure 12. Columns from left to right: Northeast, NE (a,f); Southeast, SE (b, g) ; Mountain, MT (c,h); California (d,i); Pacific Northwest, PNW (e,j). DBF and ENF PFTs are shown in the top row (a-e) and grass and ragweed PFTs are shown in the bottom row (f-j). Box and whiskers from bottom to top represent the minimum, lower quartile, median, upper quartile, and maximum. Maxima that are not visible in panels b, c and e are 1,177 grains m$^{-3}$, 1,233 grains m$^{-3}$, and 766 gains m$^{-3}$ respectively. All y-axes are the same scale for each row.

Matthew Wozniak 8/17/2017 8:51 PM

Matthew Wozniak 8/17/2017 8:51 PM

| Taxon or PFT | P $10^7$ grains m$^{-2}$ year$^{-1}$ | Reference for P | sDOY (slope/R$^2$) days °C$^{-1}$ | eDOY (slope/R$^2$) days °C$^{-1}$ |
|---|---|---|---|---|
| **Deciduous Broadleaf Forest (DBF)** | | | | |
| *Acer* | 89.1 | Tormo Molina et al. 1996 | -1.78/0.15 | -1.56/0.06 |
| *Alnus* | 210 | Helbig et al. 2004 | -8.82/0.46 | -4.88/0.26 |
| *Betula* | 140 | Jato et al. 2007 | -3.46/0.54 | -3.45/0.35 |
| *Fraxinus* | 45.1 | Tormo Molina et al. 1996 | -4.69/0.50 | -2.92/0.32 |
| *Morus* | 10 | N/A | -4.00/0.53 | -2.97/0.29 |
| *Platanus* | 121 | Tormo Molina et al. 1996 | -4.47/0.40 | -2.65/0.20 |
| *Populus* | 24.2 | Tormo Molina et al. 1996 | -2.23/0.24 | -0.31/<0.01 |
| *Quercus* | 78 | Tormo Molina et al. 1996 | -4.09/0.53 | -2.03/0.19 |
| *Ulmus (early,late)* | 3.55 | Tormo Molina et al. 1996 | -4.61/0.59, 3.06/0.12 | -2.37/0.16, 5.12/0.29 |
| DBF | 80.1 | | -4.55/0.46 | -1.94/0.13 |
| **Evergreen Needleleaf Forest (ENF)** | | | | |
| Cupressaceae | 363 | Hidalgo et al. 1999 | -5.67/0.48 | -2.67/0.17 |
| Pinaceae | 22.2 | Tormo Molina et al. 1996 | -5.72/0.45 | -5.03/0.41 |
| ENF | 193 | | -5.95/0.40 | -4.96/0.33 |
| **Grasses (GRA)** | | | | |
| Poaceae (C$_3$,C$_4$) | 8.5, 0.85 | Prieto-Baena et al. 2003 | -4.76/0.48, 0.05/<0.01 | -1.08/0.04, 2.96/0.32 |
| **Ragweed (RAG)** | | | | |
| *Ambrosia* | 119[a] | Fumanal et al. 2007 | 1.08/0.08 | 3.42/0.37 |

[a]*Ambrosia* production factor in $10^7$ grains plant$^{-1}$

**Table 1:** Production factors (P) and phenological regression coefficients for the start day of year (sDOY) and end day of year (eDOY) as a function of temperature for the 13 individual pollen-producing taxa. Individual taxa and families are organized into the four PFTs, with the two aggregated tree PFTs denoted as DBF and ENF. Regression slope (days/°C) and coefficient of determination are provided for both sDOY and eDOY (slope/$R^2$).

Allison Steiner 8/21/2017 12:43 PM
**Comment [2]:** No longer shaded… did you mean to exclude? Either add back in or modify the caption.

Matthew Wozniak 8/23/2017 11:40 AM

Allison Steiner 8/21/2017 12:43 PM

| Land cover class | NE | SE | MT | CA | PNW |
|---|---|---|---|---|---|
| *Acer* | 6.79E+04 | 2.88E+04 | 1.89E+03 | 1.97E+02 | 3.09E+03 |
| *Alnus* | 3.37E+00 | 1.23E-01 | 6.49E+01 | 1.71E+02 | 9.56E+03 |
| *Betula* | 2.99E+04 | 2.68E+03 | 2.78E+02 | 2.64E+00 | 4.82E+02 |
| *Fraxinus* | 3.96E+04 | 1.10E+04 | 3.14E+03 | 3.94E+01 | 2.76E+02 |
| *Morus* | 3.99E+03 | 2.25E+03 | 3.89E+01 | 0.00E+00 | 0.00E+00 |
| *Platanus* | 3.18E+03 | 3.38E+03 | 1.33E+01 | 1.44E+02 | 0.00E+00 |
| *Populus* | 5.48E+04 | 1.23E+03 | 4.37E+04 | 1.96E+02 | 1.55E+03 |
| *Quercus* | 1.30E+05 | 2.25E+05 | 2.51E+04 | 2.82E+04 | 1.40E+04 |
| *Ulmus* | 4.96E+04 | 2.81E+04 | 1.37E+03 | 0.00E+00 | 0.00E+00 |
| BELD DBF | 3.79E+05 | 3.03E+05 | 7.56E+04 | 2.90E+04 | 2.90E+04 |
| CLM DBF | 6.67E+05 | 4.03E+05 | 1.72E+05 | 7.93E+03 | 4.18E+04 |
| Cupressaceae | 1.85E+04 | 2.11E+04 | 7.84E+04 | 9.64E+03 | 2.35E+04 |
| Pinaceae | 8.34E+04 | 1.58E+05 | 1.79E+05 | 2.95E+04 | 1.10E+05 |
| BELD ENF | 1.02E+05 | 1.79E+05 | 2.58E+05 | 3.91E+04 | 1.34E+05 |
| CLM ENF | 1.44E+06 | 4.26E+05 | 4.66E+05 | 4.57E+04 | 5.34E+05 |

**Table 2:** Total spatial coverage (km$^2$) of tree taxa and PFTs from BELD and CLM land cover datasets in the 5 U.S. subregions (Northeast, NE; Southeast, SE; Mountain, MT; California; Pacific Northwest, PNW). All individual tree taxa are from the BELD database. BELD DBF and ENF land cover are the sums of the land cover of the taxa belonging to each PFT.

[Figure]

**Figure S1.** Linear regressions of the phenological relationship between start day-of-year (sDOY) and previous year annual average temperature (PYAAT) for all taxa. Each point signifies one station per year for pollen count data from 2003-2010 (total denoted as N). (j) ulmus_2 denotes a second (later) pollen peak in

*Ulmus* and (o) poac_2 denotes a second (later) pollen peak in Poaceae.

[Figure]

**Figure S2.** Same as Figure S1, but for end day-of-year (eDOY) versus PYAAT.

| No. | City, State | Latitude | Longitude | Years | Acknowledgements |
|---|---|---|---|---|---|
| **Northeast (38)** | | | | | |
| 1 | Albany, NY | 42.67 | -73.80 | 2003-2010 | David Shulan, M.D.
Certified Allergy Consultants
Albany, NY
Certified Allergy Consultants |
| 2 | Armonk, NY | 41.13 | -73.71 | 2003-2010 | Guy Robinson, PhD
The Louis Calder Center
Armonk, NY |
| 3 | Baltimore, MD | 39.30 | -76.61 | 2003-2010 | Jonathon Matz, MD FAAAAI &
David Golden, MD FAAAAI
Dr. Golden and Dr. Matz, LLC
Baltimore, MD |
| 4 | Boston, MA | 42.35 | -71.06 | 2010 | Immunology Research Institute of New England
Lawrence M. DuBuske, MD FAAAAI
Boston, MA |

| 5 | Brooklyn, NY | 40.65 | -73.95 | 2004-2010 | Clifford W. Bassett, MD & Mehdi Vesaghi, MD Long Island College Hospital Brooklyn, NY |
|---|---|---|---|---|---|
| 6 | Chelmsford, MA | 42.60 | -71.37 | 2003-2005 | Julian Melamed, M.D. Chelmsford, MA |
| 7 | Cherry Hill, NJ | 39.90 | -75.00 | 2003-2010 | Larchment Medical Center II Donald J. Dvorin, MD FAAAAI Cherry Hill, NJ |
| 8 | Chicago, IL | 41.84 | -87.68 | 2003-2010 | John Shane, PhD McCrone Research Institute Chicago, IL |
| 9 | Dayton, OH | 39.78 | -84.20 | 2003-2010 | Mr. Andy Roth RAPCA Dayton, OH |
| 10* | Erie, Pennsylvania | 42.13 | -80.09 | 2003-2007, 2009-2010 | Philip E. Gallagher, MD FAAAAI Allergy & Asthma Associates of Northeastern PA Erie, PA |

| 11 | Hamilton, Ontario | 43.25 | -79.1 | 2003-2005 | Jason A. Ohayan, MD Hamilton, ON |
|----|-------------------|-------|-------|-----------|----------------------------------|
| 12 | Indianapolis, IN | 39.78 | -86.15 | 2003-2009 | L.Y. Frank Wu, M.D. St. Vincent Professor Building Indianapolis, IN |
| 13 | Kansas City, MO | 39.12 | -94.55 | 2003-2007, 2009-2010 | Jay Portnoy, MD FAAAAI Children's Mercy Hospital Kansas City, MO |
| 14 | La Crosse, WI | 43.83 | -91.23 | 2003-2010 | N/A |
| 15 | Lexington, KY | 38.043 | -84.46 | 2003-2006,2008-2010 | Beth Miller,  M.D. University  of  Kentucky  Asthma  Allergy  & Immunology Lexington, KY |
| 16 | Lincoln, NE | 40.82 | -96.69 | 2004-2009 | Fred Keichel, MD Allergy, Asthma & Immunology Associates Lincoln, NE |

| 17 | London, ON | 42.98 | -81.25 | 2003-2005 | James Anderson, MLT
OSHTECH
London, ON |
|----|------------|-------|--------|-----------|----------------------------------------------|
| 18 | Louisville, KY | 38.22 | -85.74 | 2003-2010 | James L. Sublett, MD FAAAAI
Family Allergy & Asthma
Louisville, KY |
| 19 | Madison, WI | 43.08 | -89.39 | 2003-2010 | Robert Bush, MD FAAAAI
UW Medical School
Madison, WI |
| 20 | Melrose Park, IL | 41.90 | -87.86 | 2003-2010 | Joseph G. Leija, MD FAAAAI
Dr. Joseph Leija
Melrose Park, IL |
| 21 | Minneapolis, MN | 44.96 | -93.27 | 2010 | Harold B. Kaiser, MD FAAAAI
Clinical Research Institute
Minneapolis, MN |
| 22 | Newark, NJ | 40.72 | -74.17 | 2003-2009 | Alan Wolff, M.D.
UMDNJ
Newark, NJ |

| 23* | New Castle, DE | 39.62 | -75.56 | 2005-2010 | Michael McDowell
Division of Air Quality, DNREC, State of Delaware
New Castle, DE |
|------|----------------|-------|--------|-----------|-------|
| 24 | New York, NY | 40.08 | -78.43 | 2003-2010 | Guy Robinson, PhD
Fordham College at Lincoln Center
New York, NY |
| 25 | Niagrara Falls, ON | 43.09 | -79.09 | 2003-2005 | Dr. Michael Alexander, MD
Niagara Falls, ON |
| 26 | Olean, NY | 42.08 | -78.43 | 2003-2010 | Fred Lewis, MD FAAAAI
Olean, NY |
| 27 | Philadelphia, PA | 40.01 | -75.13 | 2003-2007, 2010 | Donald J. Dvorin, MD FAAAAI
Allergic Disease Associates, P.C.
Philadelphia, PA |
| 28 | Pittsburgh, PA (2) | 40.44 | -79.98 | 2003-2010 | David Skoner, MD FAAAAI
Allegheny General Hospital
Pittsburgh, PA |

| 29 | Rochester, NY | 43.17 | -77.62 | 2003-2010 | Donald W. Pulver, MD FAAAAI |
| | | | | | Allergy, Asthma & Immunology of Rochester |
| | | | | | Rochester, NY |
| 30 | Salem, MA | 42.53 | -70.87 | 2007-2010 | Paul Hannaway, M.D. |
| | | | | | Salem, MA |
| 31 | St. Claire Shores, MI | 42.49 | -82.89 | 2003-2005, 2009-2010 | Andrew I. Dzul, MD |
| | | | | | Lakeshore Ear Nose & Throat Center |
| | | | | | St. Clair Shores, MI |
| 32 | St. Louis, MO | 38.64 | -90.24 | 2003,2006-2010 | Mr. Wayne Wilhelm |
| | | | | | St. Louis County Health Department |
| | | | | | Berkeley, MO (St. Louis) |
| 33 | Washington, DC | 38.91 | -77.02 | 2003-2010 | Susan E. Kosisky MA |
| | | | | | Walter Reed Army Medical Ctr. |
| | | | | | Washington, DC |
| 34 | Waturbury, CT | 41.56 | -73.04 | 2003-2010 | Christopher Randolph, MD FAAAAI |
| | | | | | Waterbury, CT |

| 35 | Waukesha, WI | 43.01 | -88.24 | 2007-2009 | Walter Brummund, MD, PhD, FAAAAI
Allergy & Asthma Centers, S.C.
Waukesha, WI |
|----|--------------|-------|--------|-----------|------------------------------------------------------------------------------------|
| 36 | Wauwatosa, WI | 43.06 | -88.03 | 2003-2006 | Walter Brummund, MD, PhD, FAAAAI
Allergy & Asthma Centers, S.C.
Waukesha, WI |
| 37 | York, PA | 39.96 | -76.73 | 2003-2010 | Michael S. Nickels, MD PhD
Allergy and Asthma Consultants, Inc.
York, PA |

**Southeast (33)**

| 1 | Atlanta, GA | 33.76 | -84.42 | 2003-2010 | Santley M. Fineman, MD MBA FAAAAI
Atanta Allergy and Asthma
Marietta, GA (Atlanta) |
|---|-------------|-------|--------|-----------|-------------------------------------------------------------------------------------------|
| 2 | Austin, TX | 30.31 | -97.95 | 2003-2005 | Kim T. Hovanky, MD FAAAAI &
 Sheila M. Amar, MD FAAAAI, FACAAI
Allergy & Asthma Center of Georgetown
Austin, TX (Georgetown) |

| 3 | Baton Rouge, Louisiana | 30.45 | -91.13 | 2003-2005 | James M. Kidd III
Kidd Allergy Clinic
Baton Rouge, LA |
|---|---|---|---|---|---|
| 4 | Birmingham, AL | 33.53 | -86.80 | 2010 | Weilly Soong, MD FAAAAI
Birmingham-Southern College/Alabama Allergy & Asthma Center
Birmingham, AL |
| 5 | Charlotte, NC | 35.20 | -80.83 | 2003-2007 | John T. Klimas, MD FAAAAI
Carolina Asthma and Allergy Center
Charlotte, NC |
| 6 | College Station, TX | 30.60 | -96.31 | 2003-2010 | David R. Weldon, MD FAAAAI, FACAAI
Scott & White Clinic
College Station, TX |
| 7 | Corpus Christi, TX | 27.69 | -97.29 | 2003-2005 | Gary L. Smith, M.D.
Corpus Christi, TX |
| 8 | Dallas, TX | 32.79 | -96.77 | 2003-2010 | Jeffrey Adelglass, M.D.
Dr. Jeffrey Adelglass
Dallas, TX |

| 9 | Fargo, ND | 46.88 | -96.82 | 2003-2010 | Dan Dalan, MD FAAAAI
Allergy & Asthma Care Center
Fargo, ND |
|---|---|---|---|---|---|
| 10 | Flower Mound, TX | 33.03 | -97.09 | 2006-2010 | Marie H Fitzgerald, MD
North Texas Pollen Station
Flower Mound, TX |
| 11 | Fort Smith, AK | 35.37 | -94.38 | 2003-2006 | N/A |
| 12 | Georgetown, TX | 30.65 | -97.69 | 2006-2010 | Kim T. Hovanky, MD FAAAAI &
 Sheila M. Amar, MD FAAAAI, FACAAI
Allergy & Asthma Center of Georgetown
Austin, TX (Georgetown) |
| 13 | Greenville, SC | 34.84 | -82.37 | 2003-2010 | Neil L Kao MD FAAAAI
Allergic Disease and Asthma Center
Greenville, SC |
| 14 | Houston, TX | 29.77 | -95.39 | 2005-2010 | Mr. Tony Huynh
City of Houston
Houston, TX |

| 15 | Huntsville, AL | 34.71 | -86.63 | 2003-2010 | Ms. Debra Hopson
Natural Resources & Environmental Management
Huntsville, AL |
|----|---------------|-------|--------|-----------|---|
| 16 | Knoxville, TN | 35.97 | -83.95 | 2003-2010 | Michael Miller, MD FAAAAI
Allergy, Asthma and Immunology
Knoxville, TN |
| 17 | Little Rock, AK | 34.72 | -92.35 | 2004-2010 | Karl V Sitz, MD
Little Rock Allergy & Asthma Clinic
Little Rock, AR |
| 18 | Miami, FL | 25.78 | -80.21 | 2003-2005 | Elene Ubals, MD &
Richard Schiff, MD, PhD
Miami, FL |
| 19 | New Orleans, LA | 30.07 | -89.93 | 2008-1010 | W Edward Davis MD MS MBA MMM
Ochsner Clinic Foundation
New Orleans, LA |
| 20 | Ocala, FL | 29.19 | -82.13 | 2003-2005 | Dr. Karl M Altenburger
Allergy and Asthma Care of Florida
Ocala, FL |

| 21 | Oklahoma City, OK (2) | 35.47 | -97.51 | 2003-2010 | Warren V. Filley, MD FAAAAI
OK Allergy Asthma Clinic, Inc.
Oklahoma City, OK |
|----|----|----|----|----|----|
| 22 | Omaha, NE | 41.26 | -96.01 | 2003-2010 | N/A |
| 23 | Orlando, FL | 28.51 | -81.37 | 2006-2007 | Bruce A. Hornberger, MD FAAAAI
Allergy & Asthma Center of East Orlando
Orlando, FL |
| 24 | Oxford, AL | 33.61 | -85.84 | 2010 | Robert Grubbe, MD
Allergy & Asthma Center, LLC
Oxford, AL |
| 25 | Rogers, AK | 36.33 | -94.13 | 2006-2010 | Curtis L. Hedberg, MD FAAAAI
Hedberg Allergy & Asthma Center
Rogers, AR (Fort Smith) |
| 26* | Sarasota, FL | 27.34 | -82.53 | 2003-2010 | Mary Jelks, MD FAAAAI
Sarasota, FL |

| 27 | Savannah, GA | 32.02 | -81.13 | 2008-2010 | Brad H. Goodman, M.D. & Bruce D. Finkel, M.D. Coastal Allergy & Asthma, P.C. Savannah, GA |
|----|--------------|-------|--------|-----------|---|
| 28 | Tallahasseee, FL | 30.46 | -84.28 | 2003-2005 | Ronald Saff, M.D. Tallahassee, FL |
| 29 | Tampa, FL | 27.96 | -82.48 | 2003-2010 | Richard Lockey, MD FAAAAI University of South Florida Tampa, FL |
| 30 | Tulsa, OK | 36.13 | -95.92 | 2003-2010 | Estelle Levetin, PhD FAAAAI University of Tulsa Tulsa, OK |
| 31 | Waco, TX (2) | 31.57 | -97.18 | 2003-2010 | N.J. Amar, MD FAAAAI Allergy and Asthma Center Waco, TX |

**Mountain (8)**

| 1 | Bismarck, ND | 46.81 | -100.77 | 2003-2005, 2010 | Dan Dalan, M.D.
North Dakota DOH East Lab
Bismarck, ND |
|---|---|---|---|---|---|
| 2 | Colorado Springs (2) | 38.86 | -104.76 | 2003-2010 | William Storms, MD FAAAAI
The William Storms Allergy Clinic
Colorado Springs, CO |
| 3 | Draper, UT | 40.52 | -111.86 | 2009-2010 | Duane J. Harris, MD FAAAAI
Intermountain Allergy & Asthma Clinic
Draper, UT |
| 4* | Missoula, MT | 46.87 | -114.01 | 2006-2007, 2009-2010 | Emily Weiler
U of Montana, Ctr for Environmental Health Sciences
Missoula, MT |
| 5 | Salt Lake City, UT | 40.78 | -111.93 | 2003-2004, 2008 | N/A |
| 6 | Scottsdale, AZ | 33.69 | -111.87 | 2006-2007 | Michael E. Manning, M.D.
Scottsdale, AZ |

| 7* | Twin Falls, ID | 42.56 | -114.46 | 2003-2010 | N/A |
|---|---|---|---|---|---|

**California (13)**

| 1 | La Jolla, CA | 32.84 | -117.26 | 2003-2010 | Robert Reid, Jr, MD
Scripps Memorial Hospital
LaJolla, CA |
|---|---|---|---|---|---|
| 2 | Orange, CA | 33.81 | -117.82 | 2003-2010 | Sherwin A. Gillman, M.D.
Children's Hospital of Orange County-Pediatric Subspecialty Faculty
Orange, CA |
| 3 | Pasadena, CA | 34.16 | -118.14 | 2004-2005, 2010 | Philip Taylor, PhD
The Pollen Group
Pasadena, CA |
| 4 | Pleasanton, CA | 37.67 | -121.89 | 2003-2010 | N/A |

| 5 | Reno, NV | 39.54 | -119.82 | 2003-2006, 2009-2010 | Leonard Shapiro, MD FAAAAI
Allergy & Asthma Associates
Sparks, NV |
|---|---|---|---|---|---|
| 6 | Roseville, CA | 38.76 | -121.29 | 2006-2010 | Sunil P. Perera, MD FAAAAI
Allergy Medical Group of the North Area
Roseville, CA (Sacremento) |
| 7 | Sacramento, CA | 38.57 | -121.47 | 2003-2006 | Sunil P. Perera, MD FAAAAI
Allergy Medical Group of the North Area
Roseville, CA (Sacremento) |
| 8 | Salinas, CA | 36.68 | -121.64 | 2003 | Steven S. Prager, M.D.
Salinas Allergy Clinic
Salinas, CA |
| 9 | San Diego, CA | 32.82 | -117.14 | 2006, 2008-2010 | Robert T. Reid, MD
Erik and Ese Banck Clinical Research Center
San Diego, CA |
| 10 | San Jose, CA (2) | 37.30 | -121.85 | 2003-2010 | Theodore Chu, MD FAAAAI
San Jose, CA |

| 11 | Santa Barbara, CA | 34.43 | -119.72 | 2003-2010 | Myron Liebhaber, M.D. Sansum - Santa Barbara Medical Foundation Clinic Santa Barbara, CA |
|---|---|---|---|---|---|
| 12 | Stockton, CA | 37.97 | -121.31 | 2009-2010 | Gregory W. Bensch, MD FAAAAI & George W Bensch, MD FAAAAI Allergy, Immunology and Asthma Medical Group Stockton, CA |

**Pacific NW (4)**

| 1 | Crescent City, CA | 41.76 | -124.20 | 2009-2010 | N/A |
|---|---|---|---|---|---|
| 2* | Eugene, OR | 44.05 | -123.11 | 2003-2010 | Kraig W. Jacobson, MD, FAAAAI Allergy & Asthma Research Group Eugene, OR |
| 3 | Seattle, WA | 47.62 | -122.35 | 2003-2010 | N/A |

| 4* | Vancouver, WA | 45.63 | -122.64 | 2003-2010 | Raymond Brady, M.D. and Joseph Hassett, M.D. Vancouver, WA |
|---|---|---|---|---|---|

**Table S1.** AAAAI pollen counting stations included in development and evaluation of model. Categorized by U.S. subregions Northeast, Southeast, Mountain, California, and Pacific Northwest. Sites equipped with two samplers are denoted in parentheses (ex. Pittsburgh, PA (2)). Station numbers with asterisk (*) are collocated with DayMet temperature for phenological regressions.

| Taxon | Maximum $P_{avgmax}$ (grains m$^{-3}$) | Average $P_{avgmax}$ (grains m$^{-3}$) |
|---|---|---|
| *Acer* | 535 | 212 |
| *Alnus* | 176 | 73 |
| *Ambrosia* | 379 | 193 |
| Arecaceae | 27 | 13 |
| *Artemisia* | 76 | 36 |
| Asteraceae | 64 | 28 |
| *Betula* | 564 | 232 |
| *Carpinus/Ostrya* | 71 | 29 |
| *Carya* | 126 | 54 |
| *Celtis* | 155 | 65 |
| Chenopodiaceae/Amaranthaceae | 80 | 29 |
| *Corylus* | 13 | 5 |
| Cupressaceae | 1949 | 960 |
| Cyperaceae | 37 | 12 |
| *Eupatorium* | 27 | 13 |
| *Fagus* | 66 | 17 |
| *Fraxinus* | 462 | 227 |
| Graminae / Poaceae | 197 | 98 |
| *Juglans* | 71 | 34 |
| *Ligustrum* | 13 | 6 |
| *Liquidambar* | 109 | 44 |
| *Morus* | 940 | 398 |
| *Myrica* | 50 | 25 |
| *Olea* | 197 | 43 |
| Other Grass Pollen | 85 | 42 |
| Other Tree Pollen | 863 | 332 |
| Other Weed Pollen | 160 | 51 |
| Pinaceae | 645 | 328 |
| *Plantago* | 33 | 13 |
| *Platanus* | 191 | 103 |
| *Populus* | 367 | 158 |
| *Prosopis* | 34 | 16 |
| *Pseudotsuga* | 7 | 3 |

| | | |
|---|---|---|
| *Quercus* | 2106 | 1081 |
| *Rumex* | 59 | 23 |
| *Salix* | 123 | 58 |
| *Tilia* | 19 | 6 |
| *Tsuga* | 12 | 5 |
| *Typha* | 19 | 10 |
| *Ulmus* | 556 | 249 |
| **Unidentified Pollen** | 198 | 57 |
| **Urticaceae** | 86 | 36 |

**Table S2.** *Maximum and average $P_{avgmax}$* used for selecting pollen taxa for this study. Shaded taxa are selected taxa based on minimum of 1) 100 grains m$^{-3}$ for the maximum $P_{avgmax}$ and 2) 70 grains m$^{-3}$ for the average $P_{avgmax}$.

Matthew Wozniak 8/9/2017 11:27 AM

---

## Author Comment (AC2) · 31 Aug 2017

Response to Referee #2

Thank you for the constructive suggestions to improve the manuscript. Line numbers have changed in the manuscript mark-up below; thus, line number references in the author responses are to new line numbers generated because of additional or rearranged content. Referee #2's comments are individually listed below with a corresponding author response.

Comment: Line 28: Wind-borne pollen diameters can range more than 70 $\mu$m.

Response: Agreed. We have updated this line to paraphrase pollen size ranges described in the literature: "ranging typically from 15 to 60 $\mu$m in diameter, while sometimes exceeding 100 $\mu$m (Cecchi 2014; Sofiev et al. 2014)" on Line 30.

Comment: Line 221: BELD is not based only on land surveys. The authors should revise and include version information. Response: Thank you for bringing this to our attention. The description of BELD has been updated for accuracy: "The Biogenic Emissions Landuse Database version 3 (BELD) provides vegetation species distributions at 1 km resolution over the continental United States based on satellite imagery, aerial photography and ground surveys, as well as other land cover classification data such as geographical boundaries (Kinnee et al. 1997; https://www.epa.gov/air-emissions-modeling/biogenic-emissions-landuse-database-version-3-beld3)." (Lines 262-266). Access to the version used in this manuscript is now added as a link to the data access webpage with the citation.

Comment: Section 4.3 needs enhancements to better explain how the production factor was obtained for each modeled taxon due to non-uniform methodology. Furthermore, Table 2 does not exist.

Response: We apologize for the table reference error – the original reference in Section 4.3 was meant to refer to Table 1, which is included in the revised mark-up.

Section 4.3 has been modified to describe the production factor implementation in greater detail. The section now reads as follows: "Annual production factors (grains m-2 year-1; or grains stem-1 year-1 for ragweed) for each modeled taxon are provided in Table 1. The annual pollen production factor (pannual) defines the amount of pollen produced per vegetation biomass per year based on literature values. Tormo Molina et al. (1996) report the annual pollen productivity in grains tree-1 year-1 measured from a number of representative trees from several taxa. Morus has no known reference for production factor and was assumed to be 10x107 grains m-2 year-1, conservatively at the low end of the range for other deciduous broadleaf taxa. Other tree taxa and grasses are reported in grains m-2 year-1, while ragweed is reported in grains stem-1 year-1 (Helbig et al. 2004; Jato, Rodríguez-Rajo, and Aira 2007; Hidalgo, Galán, and

Domínguez 1999; Prieto-Baena et al. 2003; Fumanal, Chauvel, and Bretagnolle 2007). To convert from grains tree-1 year-1 to grains m-2 year-1, the production factors are multiplied by the tree crown area given in Table II of Tormo Molina et al. (1996) After sensitivity experiments of running pollen emissions in RegCM4, we find that the litera-ture value of pannual for Poaceae provides better agreement with observations for C4 grass when reduced by a factor of 10, thus we use this value. To obtain the coefficient of daily pollen production over the duration of the phenological curve, ðİŻ¿phen, the integral of the daily pollen production is normalized to pannual as demonstrated by Equation 2."

Comment: Section 5. The regional climate model setup needs to be described in more detail (i.e. number of cells, resolution, vertical structure, etc.).

Response: Lines have been added in Section 5 to address this deficit: "The pollen tracer transport scheme is extended from one to four bins in this study to simulate the four PFTs (DBF, ENF, GRA, and RAG), with tracer bin particle effective diameters of 28 $\mu$m, 40 $\mu$m, 35 $\mu$m and 20$\mu$m, respectively. Additionally, the temporal emissions input is updated to accommodate daily pollen emissions (grains m-2 day-1)." (Lines 500-503). "The horizontal resolution is 25-km with 144x243 grid cells on a Lambert Conformal Projection centered on 39°N, 100°W with parallels at 30°N and 60°N (Figure 1). The vertical resolution includes 18 vertical sigma levels. Boundary conditions are driven by ERA-Interim Reanalysis while sea surface temperatures are prescribed from NOAA Optimum Interpolation SSTs (Dee et al. 2011; Smith et al. 2008)." (Lines 507-511).

Comment: A taxa-based database comparison providing spatial coverage values for each region would be a useful addition.

Response: To address this concern, we have included a new table to display the total land cover for each tree taxon and PFT in each U.S. subregion (Table 2). This provides a useful comparison of regional land cover when comparing the relative magnitudes of pollen emissions and pollen counts shown in Figures 6-13. This table is now introduced in Section 3.1 on Line 277. An additional reference is made on Line 292 in the sentence, "Overall, the CLM4 land cover fractions for forest PFTs are higher on aver-age than the summed BELD taxa, about 2 to 10 times as much in each region, with the exception of California subregion DBF where CLM4 landcover is about half of that in the BELD dataset (Table 2)."

Comment: References need to be carefully checked - i.e. Zhang, R. et al. (2014)

Response: All references have now been carefully checked against their articles, as is noted in the author response to Referee #1, and corrections were made to any refer-ences with errors or missing components. We apologize for inconveniences caused by errors in the referencing.

Comment: Production factors and the units listed in Table 1 must be properly refer-enced.

Response: References have been added as a column in Table 1 to the source in-formation for creating the model production factors. Units are included in the column titles.

Best,

Matthew Wozniak

Please also note the supplement to this comment:
https://www.geosci-model-dev-discuss.net/gmd-2017-105/gmd-2017-105-AC2-supplement.pdf